# Training-Free Guided Diffusion for Planning:
# A Unified Framework via Doob's $h$-Transform with Safety Guarantees

**Kenta Hoshino** [1 2]   **Yashaswi Shashank Aluru** [* 3]   **Xiyu Deng** [* 3]   **Yorie Nakahira** [3]

## Abstract

This paper studies the theoretical foundations of guidance mechanisms in continuous-time score-based diffusion models. We adopt Doob's $h$-transform as a principled framework for characterizing ideal guided diffusion processes and analyze the discrepancy between ideal and approximate guidance. Our analysis provides explicit error bounds and yields probabilistic guarantees on satisfying prescribed constraints, which are particularly important for safety-critical planning. We further show that the Doob-based formulation induces a stochastic optimal control problem, enabling practical guidance design without additional model training. We demonstrate the effectiveness of the proposed framework on robotic navigation tasks, including language-conditioned planning.

## 1. Introduction

Robotic systems are increasingly expected to plan in complex environments under task- and environment-dependent constraints. Meeting these demands requires principled planning methods that provide reliability, such as guarantees on goal achievement and constraint satisfaction, while remaining adaptable across tasks and environments. Diffusion models have recently emerged as a powerful framework for generative modeling (Sohl-Dickstein et al., 2015; Ho et al., 2020; Song et al., 2021) and have shown strong performance in robotic planning (Janner et al., 2022; Chi et al., 2025), as they can represent complex, multimodal trajectory distributions and enable conditional generation via guidance mechanisms (Dhariwal & Nichol, 2021; Ho & Salimans).

*Equal contribution   [1]School of Computing, Institute of Science Tokyo, Tokyo, Japan [2]Denso IT Laboratory, Tokyo, Japan [3]Electrical and Computer Engineering, Carnegie Mellon University, Pittsburgh, USA. Correspondence to: Kenta Hoshino <hoshino@comp.isct.ac.jp>, Yorie Nakahira <ynakahir@andrew.cmu.edu>.

*Proceedings of the $43^{rd}$ International Conference on Machine Learning*, Seoul, South Korea. PMLR 306, 2026. Copyright 2026 by the author(s).

Recent work has applied guidance-based diffusion to decision making and planning by generating trajectories conditioned on start/goal specifications, costs or returns, or auxiliary signals (Janner et al., 2022; Zhong et al., 2023; Ziyuan et al., 2023; Zhou et al., 2025). However, existing methods usually rely on heuristic or learning-based guidance, making it difficult to provide guarantees on goal achievement or constraint satisfaction. In safety-critical scenarios such as robotic planning, theoretical guarantees—for example, on the probability of success or constraint satisfaction—are essential, although relatively few existing studies explicitly discuss this perspective.

Recently, Doob's $h$-transform (Doob, 1984) has been recognized as a fundamental theoretical tool for understanding guided diffusion models, as it characterizes diffusion processes conditioned on reaching a target set (Heng et al., 2025; Du et al., 2024; Denker et al., 2024; Nguyen et al., 2025; Didi et al., 2024). While Doob's $h$-transform provides an ideal form of guidance with strong guarantees, its practical use is limited by computational intractability and the need for approximation. Moreover, its implications for quantitative reliability in planning settings remain underexplored.

In this work, we develop a theoretical framework for guided diffusion grounded in Doob's $h$-transform, with a particular focus on robotic planning. Our key contributions are (i) a quantitative analysis of the discrepancy between ideal Doob-based guidance and practical approximate guidance, and (ii) explicit bounds on the success probability of satisfying task constraints under approximate guidance. These results are derived by viewing approximate guidance as a perturbation of the ideal Doob-based guidance and by quantifying how approximation errors affect reliability in terms of both success probability and relative entropy, which is particularly important in safety-critical planning applications.

We further apply the Doob's $h$-transform-based framework to practical approximation through an equivalent formulation in terms of stochastic optimal control. Specifically, we show that the guidance design induced by Doob's $h$-transform admits an equivalent characterization as a stochastic optimal control problem. This equivalence allows us to leverage established tools from stochastic optimal con-

trol theory to compute guidance functions by solving the corresponding control problem.

An important consequence of this formulation is that guidance design can be reframed as the design of a terminal cost function in the associated optimal control problem. Combined with the theoretical probability bounds derived in this paper, this perspective provides a principled guideline: better-designed terminal cost functions lead to higher success probabilities in satisfying task constraints during planning. Additionally, from this perspective, efficient guidance design primarily reduces to the construction of appropriate terminal cost functions in the stochastic optimal control formulation. This also enables us to employ practical techniques from stochastic optimal control theory, such as iterative linear quadratic Gaussian (iLQG) and path integral control, enabling task-adaptive guidance without retraining diffusion models for each task.

Finally, we demonstrate in simulation that the proposed method achieves improved performance, safety, and generalization with reduced online computational cost. We further discuss language-based guidance from the perspective of the proposed framework, showing that task specifications expressed in natural language can be naturally incorporated by designing appropriate terminal cost functions.

Our contributions are summarized as follows:

- We develop a principled framework for guided diffusion grounded in Doob's $h$-transform, enabling constraint-aware guidance with theoretical guarantees while requiring little or no task-specific retraining.

- We show that the resulting guidance design admits an equivalent formulation as a stochastic optimal control problem, and demonstrate that practical guidance functions can be computed efficiently using path integral control and related optimal control techniques.

- We empirically validate the proposed framework on robotic planning tasks. The results demonstrate computational efficiency (Figure 2), generalization (Figure 3), and reliable performance (Table 1).

## 2. Related Works

We provide a more detailed discussion of related work in Appendix B.

*Theoretical studies on diffusion models.* Existing theoretical analyses of diffusion models have primarily focused on convergence and sampling properties for *unguided* dynamics, often quantified via total variation or Wasserstein-type metrics, among others (Chen et al., 2023; Tang & Zhao, 2024; De Bortoli et al., 2021); see also (Tang & Zhao, 2025) for an overview. These results, however, do not directly carry over to guided diffusion, especially the evaluation of the success probability of guided diffusion.

*Diffusion and stochastic optimal control.* Recent work has begun to connect diffusion models with stochastic optimal control. Several studies interpret reverse-time diffusion dynamics as optimal control problems (Li & Pereira, 2024; Zhang & Chen, 2022; Behjoo & Chertkov, 2025). Related ideas have also been explored for training-free guidance in diffusion models (Kim et al., 2025; Huang et al., 2024) and for using diffusion models to improve MPPI-style planning (Huang & Liu, 2024). Unlike these works, we focus on the relation between guided diffusion models and stochastic optimal control via Doob's $h$-transform.

*Diffusion with constraints.* Several recent studies have incorporated constraints directly into diffusion dynamics or inference procedures, including reflected or barrier-based diffusion (Lou & Ermon, 2023; Fishman et al., 2023b), extensions to Riemannian manifolds (Fishman et al., 2023a), projection-based methods, constrained distributional optimization (Khalafi et al., 2024), and dual-space constructions such as mirror diffusion (Liu et al., 2023). While these approaches enforce constraints at the level of sampling or training, they typically require additional task-specific training. Avoiding such retraining is particularly important in robotic planning.

*Diffusion for planning.* Diffusion models have been applied to planning by modeling trajectory distributions and sampling plans conditioned on goals, costs, or returns. Diffuser demonstrates trajectory diffusion as a flexible offline planner (Janner et al., 2022), and subsequent work improves efficiency via distillation or acceleration (Lu et al., 2025). Several studies incorporate costs or constraints into diffusion-based planning, including ensemble-of-costs guidance (Saha et al., 2024), potential-based diffusion motion planning (Luo et al., 2024), and diffusion-generated seeds for trajectory optimization (Huang et al., 2025). Diffusion models have also been integrated with MPC frameworks (Zhou et al., 2025). Recent studies, such as Xiao et al. (2025), have begun to impose safety specifications or constraints within discrete-time diffusion-based planning, highlighting the need for principled analysis of continuous-time diffusion models for planning. We develop such a continuous-time framework grounded in Doob's $h$-transform, which provides a basis for safety-aware planning with explicit theoretical guarantees.

## 3. Problem Setting: Guided Diffusion Models

### 3.1. Diffusion Models

We briefly review score-based continuous-time diffusion models (Song et al., 2021; Tang & Zhao, 2025), which form the basis of our problem formulation for guided diffusion models. Additional details are provided in Appendix C.

**Notation.** We write $X \sim P$ to indicate that the random variable $X$ follows the distribution $P$, and use $\mathcal{L}(X)$ to denote the law of $X$. A complete list of notation used in this paper is provided in Appendix A.

Diffusion models generate samples from a data distribution through a forward noising process and a reverse denoising process. The forward process gradually adds noise to the data, transforming the data distribution to a simple data distribution that is easy to sample from, such as a Gaussian. The reverse process learns to approximate the time-reversal of the forward noising process, enabling generative sampling.

Let $P_{\text{data}}$ be the data distribution on $\mathbb{R}^n$. The forward noising process is given by the stochastic differential equation (SDE): for $t \in [0, T]$,

$$dX_t = f(t, X_t)dt + g(t)dW_t, \ X_0 \sim P_{\text{data}}, \quad (1)$$

where $f : [0, T] \times \mathbb{R}^n \to \mathbb{R}^n$ is the drift, $g : [0, T] \to \mathbb{R}$ is the diffusion coefficient, and $W_t$ is the $n$-dimensional standard Brownian motion. The forward SDE (1) can be intuitively understood as gradually adding noise to the data $X_0 \sim P_{\text{data}}$ so that the marginal distribution of $X_T$ becomes a simple distribution at $T$.

The reverse denoising process is modeled by the reverse-time SDE (Anderson, 1982; Haussmann & Pardoux, 1986). Let $\{Y_t\}_{t \in [0,T]}$ denote the solution to the reverse SDE, which is constructed so that $\mathcal{L}(Y_t) = \mathcal{L}(X_{T-t})$ for $t \in [0, T]$. The reverse-time SDE is given by

$$dY_t = \bar{f}(t, Y_t) \, dt + \bar{g}(t) \, d\bar{W}_t, \quad t \in [0, T], \quad (2)$$

where $\bar{W}_t$ is an $n$-dimensional standard Brownian motion, and the drift $\bar{f} : [0, T] \times \mathbb{R}^n \to \mathbb{R}^n$ and diffusion coefficient $\bar{g} : [0, T] \to \mathbb{R}$ are given by

$$\bar{f}(t, x) = -f(T - t, x) + g(T - t)^2 \nabla_x \log p(T - t, x), \quad (3)$$

$$\bar{g}(t) = g(T - t), \quad (4)$$

with $p(t, x)$ denoting the probability density function of the solution to the forward SDE (1) at time $t$.

Note that, for notational convenience, we define the reverse SDE over the time interval $t \in [0, T]$, whereas it is often formulated over $t \in [T, 0]$ in the literature.

Theoretically, the reverse SDE (2) should start with $Y_0 \sim \mathcal{L}(X_T)$ to satisfy $\mathcal{L}(Y_t) = \mathcal{L}(X_{T-t})$ for $t \in [0, T]$. However, in practice, one typically initializes the reverse process from a simple distribution that approximates $\mathcal{L}(X_T)$, such as a Gaussian distribution. We denote such a noise distribution by $P_{\text{noise}}$ throughout this paper. Accordingly, we assume that $Y_0 \sim P_{\text{noise}}$ and that $P_{\text{noise}} \approx \mathcal{L}(X_T)$ in an appropriate sense, e.g., in terms of weak convergence.

In practice, the score function $\nabla \log p(t, x)$ is generally intractable. Therefore, the reverse SDE (2) is approximated as

$$dY_t = \bar{f}_\theta(t, Y_t) \, dt + \bar{g}(t) \, d\bar{W}_t, \quad t \in [0, T], \quad (5)$$

where the drift term $\bar{f}_\theta : [0, T] \times \mathbb{R}^n \to \mathbb{R}^n$ is given by

$$\bar{f}_\theta(t, x) = -f(T - t, x) + g(T - t)^2 s_\theta(T - t, x), \quad (6)$$

and $Y_0 \sim P_{\text{noise}}$. Here, $s_\theta : [0, T] \times \mathbb{R}^n \to \mathbb{R}^n$ denotes a neural network parameterized by $\theta$ that approximates the score function $\nabla_x \log p(t, x)$.

The neural network $s_\theta$ is trained to approximate the score function using a standard score-matching objective. We provide the explicit loss function in Appendix C.

Simulating the reverse-time SDE (5) with $Y_0 \sim P_{\text{noise}}$ yields a sample $Y_T$ whose distribution approximates the data distribution $P_{\text{data}}$.

## 3.2. Guided Diffusion Models

We present the problem setting of guided diffusion models (Dhariwal & Nichol, 2021; Ho & Salimans).

In score-based diffusion models, guidance is typically implemented by modifying the reverse-time SDE (5) through the addition of a control term to the drift. The guided reverse SDE is given by

$$dY_t^g = \left( \bar{f}_\theta(t, Y_t^g) + \bar{g}(t)u(t, Y_t^g) \right) dt + \bar{g}(t)d\bar{W}_t, \quad (7)$$

for $t \in [0, T]$, where $u : [0, T] \times \mathbb{R}^n \to \mathbb{R}^n$ is the guidance function designed to steer the generated samples toward satisfying prescribed conditions.

This study focuses on the design of the guidance function $u$ in the guided reverse SDE (7), which is formulated as the following problem.

**Problem 3.1.** Consider the guided reverse SDE (7) with data distribution $P_{\text{data}}$ and noise distribution $P_{\text{noise}}$. Let $S \subset \mathbb{R}^n$ be a measurable set. The goal is to design the guidance function $u$ such that the solution $\{Y_t^g\}_{t \in [0,T]}$ to the guided reverse SDE satisfies $\mathbb{P}(Y_T^g \in S) \geq 1 - \epsilon$ for a given $\epsilon \in (0, 1)$.

Guarantees on success probability are particularly important in applications such as robotic planning, where guided diffusion models have recently been employed. In these settings, the guidance term $u$ can be designed in various ways, ranging from heuristic strategies to more principled approaches such as classifier guidance (Dhariwal & Nichol, 2021) and classifier-free guidance (Ho & Salimans). It is known that Doob's $h$-transform provides a principled way to determine the guidance term, as it guarantees that the guided diffusion process satisfies prescribed conditions, and can therefore be regarded as a theoretical ideal for guidance

design. However, its analytical form is rarely available and, in practice, its implementation typically requires learning or approximation. Throughout this work, we view Doob's $h$-transform as the ideal conditioned diffusion process, and study guided diffusion models as approximations to this ideal.

This leads to the central question of this work: *Can Doob's h-transform provide a principled framework for tractable guidance design in diffusion models, while addressing the guarantee posed in Problem 3.1? Moreover, can this framework provide useful insights into the design and interpretation of guidance functions?*

# 4. Design Principle of Guidance from Perspective of Doob's $h$-transform

This section presents our theoretical framework for designing the guidance function $u$ in the guided reverse SDE (7). We adopt Doob's $h$-transform as a theoretical ideal for guidance design. Within this framework, we study guidance functions as approximations to the Doob-optimal guidance and analyze the consequences of such approximations.

### 4.1. Guidance Design based on Doob's $h$-transform

Doob's $h$-transform provides a principled framework for constructing Markov processes conditioned on specific events, such as reaching a target set at a prescribed time (Doob, 1984; Rogers & Williams, 2000). From the perspective of guided diffusion models, this suggests a natural mechanism for designing the guidance function $u$ in the guided reverse SDE (7) so that the generated sample $Y_T^g$ satisfies the prescribed target condition.

Several recent studies have exploited this connection to develop learning-based guidance frameworks (Heng et al., 2025; Du et al., 2024; Denker et al., 2024; Nguyen et al., 2025; Didi et al., 2024). While these methods provide powerful tools for obtaining guidance in practice, a unified theoretical framework for analyzing guidance design remains to be explored.

In this work, we place Doob's $h$-transform at the center of our formulation and develop a general framework for interpreting and analyzing guidance functions as approximations to the Doob-optimal guidance.

Doob's $h$-transform provides a principled guideline for designing the guidance function in guided diffusion models. Given the guided reverse SDE (7) with a trained score network $s_\theta$, our goal is to design the guidance function $u$ so that the solution $Y_t^g$ reaches a target set $S$ at time $T$ with high probability. Following the general framework of Doob's $h$-transform reviewed in Appendix D, such a guidance can be constructed as follows.

Let $h : [0, T] \times \mathbb{R}^n \to \mathbb{R}_{\geq 0}$ be a positive function satisfying the backward Kolmogorov equation

$$\frac{\partial h}{\partial t}(t, x) + \bar{L}_t h(t, x) = 0, \tag{8}$$

where the infinitesimal generator $\bar{L}_t$ is defined by

$$\bar{L}_t \phi(x) = \nabla_x \phi(x)^\top \bar{f}_\theta(t, x) + \frac{1}{2} \bar{g}(t)^2 \Delta_x \phi(x), \tag{9}$$

for $\phi \in C^2(\mathbb{R}^n)$, with terminal condition

$$h(T, x) = \mathbf{1}_S(x), \quad x \in \mathbb{R}^n, \tag{10}$$

where $\mathbf{1}_S$ denotes the indicator function of the set $S \subset \mathbb{R}^n$. Then, the guidance function ensuring almost sure satisfaction of $Y_T^g \in S$ is given by

$$u(t, x) = \bar{g}(t) \nabla_x \log h(t, x). \tag{11}$$

This leads to the guided reverse SDE

$$dY_t^{g,h} = \left[ \bar{f}_\theta(t, Y_t^{g,h}) + \bar{g}(t)^2 \nabla_x \log h(t, Y_t^{g,h}) \right] dt + \bar{g}(t) d\bar{W}_t, \tag{12}$$

for $t \in [0, T]$. This construction provides an ideal guidance design that guarantees $Y_T^g \in S$ with probability one. The following can be shown by slightly generalizing Theorem 7.11 of (Särkkä & Solin, 2019)

**Theorem 4.1.** *Assume that there exists a sufficiently regular solution $h$ to the backward Kolmogorov equation (8) with terminal condition (10). Let $\{Y_t^{g,h}\}_{t \in [0,T]}$ be the solution to the guided reverse SDE (12), and denote its terminal law by $\mu_T^h := \mathcal{L}(Y_T^{g,h})$. Then, $\mu_T^h(S) = 1$.*

While Doob's $h$-transform provides a principled guideline for guidance design, addressing Problem 3.1, solving the backward Kolmogorov equation (8) is generally computationally expensive. Consequently, existing studies often rely on learning-based approximations inspired by Doob's $h$-transform. This motivates us to focus on the theoretical analysis of approximate guidance designs derived from the Doob framework. From this perspective, our analysis naturally extends to a broad class of guidance mechanisms and provides a unified way to relate them to Doob's $h$-transform. In addition, the use of the indicator function in the terminal condition (10) may induce singular behavior in $\nabla_x \log h$, which further motivates the use of smooth approximations in practice.

### 4.2. Theoretical Analysis of Guidance from Doob's $h$-transform

We now develop a theoretical analysis of guided diffusion models from the perspective of Doob's $h$-transform. As

discussed above, the ideal guidance induced by Doob's $h$-transform is generally intractable in practice and necessitates approximation. Our analysis focuses on quantifying the discrepancy between the ideal guided diffusion process based on Doob's $h$-transform and the approximate guided diffusion process induced by an approximate guidance function. In particular, we derive error bounds and establish guarantees on the probability of reaching the prescribed set $S$.

We begin by formalizing the approximation considered throughout this study. Specifically, we consider an approximation $\tilde{h}$ of the function $h$ in Doob's $h$-transform introduced in the previous subsection. Given such an approximate function $\tilde{h}$, the guided reverse SDE with the corresponding approximate guidance is defined as

$$dY_t^{g,\tilde{h}} = \left( \bar{f}_\theta(t, Y_t^{g,\tilde{h}}) + \bar{g}(t)^2 \nabla_x \log \tilde{h}(t, Y_t^{g,\tilde{h}}) \right) dt \\ + \bar{g}(t) d\bar{W}_t. \tag{13}$$

The primary approximation considered in this study is based on approximating the function $h$ by solving the backward Kolmogorov equation (8) with a smoothed terminal condition. Specifically, we assume that $\tilde{h}$ satisfies

$$\frac{\partial \tilde{h}}{\partial t}(t, x) + \bar{L}_t \tilde{h}(t, x) = 0, \tag{14}$$

with terminal condition

$$\tilde{h}(T, x) = \tilde{h}_T(x), \quad x \in \mathbb{R}^n, \tag{15}$$

where $\tilde{h}_T : \mathbb{R}^n \to \mathbb{R}_{\geq 0}$ is a smooth approximation of the indicator function $\mathbf{1}_S$.

The central question of this study is whether the approximate guidance can ensure satisfaction of the target condition $Y_T^{g,\tilde{h}} \in S$ with high probability, as posed in Problem 3.1. While the ideal guidance induced by Doob's $h$-transform guarantees almost sure satisfaction of the target condition, as indicated in Theorem 4.1, practical implementations necessarily rely on approximate guidance. This motivates us to analyze the discrepancy between the distributions of the solutions to the ideal guided reverse SDE (12) and the approximate guided reverse SDE (13).

Since guided diffusion models are inherently probabilistic models that aim to generate samples from a modified data distribution conditioned on the target set $S$, it is natural to quantify this discrepancy in terms of probability measures. In this work, we characterize the discrepancy using the relative entropy (Kullback–Leibler divergence) between the marginal distributions at time $T$ of the two guided processes, which correspond to the induced output distributions of the guided diffusion models. This characterization enables us to estimate the probability of reaching the target set under approximate guidance.

We assume the following condition on $h$ and $\tilde{h}$.

**Assumption 4.2.** For the guidances induced by Doob's $h$-transform using the functions $h$ and $\tilde{h}$, define

$$\rho := \\ \int_0^T \mathbb{E}_{\mathbb{P}^h} \left[ \left\| \bar{g}(t) \left( \nabla \log \tilde{h}(t, Y_t^h) - \nabla \log h(t, Y_t^h) \right) \right\|^2 \right] dt, \tag{16}$$

where $\mathbb{E}_{\mathbb{P}^h}$ denotes the expectation with respect to the probability measure induced by $\{Y_t^h\}_{t \in [0,T]}$ of the SDE (12). We assume that $\rho < \infty$.

In the following, for the ideal and approximate guided reverse SDEs (12) and (13), let $\mu_T^h$ and $\mu_T^{\tilde{h}}$ denote the marginal distributions of $Y_T^{g,h}$ and $Y_T^{g,\tilde{h}}$ at time $T$, that is, $\mu_T^h := \mathcal{L}(Y_T^{g,h})$ and $\mu_T^{\tilde{h}} := \mathcal{L}(Y_T^{g,\tilde{h}})$.

**Theorem 4.3.** *Consider the ideal guided reverse diffusion process defined by the SDE* (12)*, where the guidance is constructed via Doob's $h$-transform using a function $h$ satisfying the backward Kolmogorov equation* (8) *with terminal condition $h(T, x) = \mathbf{1}_S(x)$. Consider also the approximate guided reverse diffusion process defined by the SDE* (13)*, where the guidance is constructed using an approximate function $\tilde{h}$. Suppose that Assumption 4.2 holds. Then the relative entropy between the marginal distributions at time $T$ satisfies*

$$D_{\mathrm{KL}}(\mu_T^h \| \mu_T^{\tilde{h}}) \leq \frac{1}{2}\rho \tag{17}$$

*where $\rho$ is defined in* (16)*.*

The proof of Theorem 4.3 is provided in Appendix H.

Theorem 4.3 quantifies the discrepancy between the ideal guided diffusion process induced by Doob's $h$-transform and an approximate guided process in terms of the relative entropy between their terminal distributions. Since these terminal distributions characterize the output distributions of the guided diffusion models, this result provides a principled criterion for evaluating the quality of an approximate guidance function at the distributional level.

The above result allows us to derive a lower bound on the probability of reaching the target set under the approximate guided diffusion process, thereby addressing Problem 3.1. Note that, in the following, the success probability $\mathbb{P}(Y_T^g \in S)$ and the tolerance parameter $\epsilon$ in Problem 3.1 correspond to $\mu_T^{\tilde{h}}(S)$ and $\sqrt{\rho/2}$, respectively.

**Theorem 4.4.** *Under the same assumptions as in Theorem 4.3, if $\rho < 2$, then the success probability under the approximate guided process satisfies*

$$\mu_T^{\tilde{h}}(S) \geq 1 - \sqrt{\frac{1}{2}\rho}. \tag{18}$$

The proof of Theorem 4.4 is also provided in Appendix H.

Theorem 4.4 provides insight into the design of guidance functions from the perspective of Doob's $h$-transform. The bound (18) in Theorem 4.4 shows that the probability of satisfying the prescribed condition is explicitly controlled by the quantity $\rho$ defined in (16). In particular, if the approximate guidance in (13) is close to the ideal Doob guidance in an $L^2$ sense along paths, then the guided diffusion process reaches the target set with high probability. This observation is especially relevant in applications such as robotic planning, where constraint satisfaction is a critical requirement. When aiming to ensure a desired level of success probability, Theorem 4.4 provides a principled criterion for guidance design in terms of $\rho$.

The viewpoint presented here naturally connects a wide class of existing guidance strategies, including learning-based and heuristic methods, to a unified theoretical framework centered on Doob's $h$-transform.

*Remark* 4.5. The terminal condition $h(T, x) = \mathbf{1}_S(x)$ is discontinuous and may not yield a classical solution to the backward Kolmogorov equation. Throughout the analysis, we interpret this setting as the limit of smooth terminal functions approximating $\mathbf{1}_S$. Accordingly, Theorem 4.4 should be understood as characterizing this limiting behavior.

**Scope of the theoretical guarantees.** The guarantees in Theorems 4.3 and 4.4 focus on the analysis of the guidance approximation effect. In these theorems, we consider the learned reverse-time diffusion model (5) with the trained score network $s_\theta$. The above discussion assumes that the ideal and approximate guided processes are built on this learned base process, where the ideal guidance induced by Doob's $h$-transform is replaced with an approximate guidance. Thus, Theorem 4.3 quantifies the effect of guidance approximation in terms of the discrepancy between the terminal distributions of the ideal and approximate guided processes, while treating the learned reverse-time diffusion model as given. Additionally, Theorem 4.4 provides a lower bound on the probability of satisfying the target constraint under approximate guidance in the same setting.

The discussion above does not include other sources of approximation error, such as score approximation error or time-discretization error. These errors are orthogonal to the guidance-design error studied here. For example, Chen et al. (2023) analyze score approximation and time-discretization errors in score-based diffusion models using KL divergence. Since our analysis also uses KL divergence to quantify the error of approximate guidance, it is compatible with such analyses and can in principle be combined with them to obtain a total error estimate and probabilistic guarantees for satisfying the target constraint. See also Appendix I for additional discussions.

## 5. A Unified View of Guidance Design under the Doob Framework

This section presents a unified perspective on guidance design approaches based on stochastic optimal control from the viewpoint of Doob's $h$-transform. Under the stochastic optimal control formulation, guidance functions in diffusion models are naturally interpreted as optimal controls.

The following result formalizes this connection by showing that the guidance design based on Doob's $h$-transform can be cast as a stochastic optimal control problem, which allows us to leverage tools from stochastic optimal control theory to efficiently compute or approximate the guidance function.

We consider the following stochastic optimal control problem (see Appendix E for details). The objective is to minimize the cost functional

$$J(u) = \mathbb{E}\left[\int_0^T \frac{1}{2}\|u(t, Y_t^u)\|^2\, dt - \log \tilde{h}_T(Y_T^u)\right], \quad (19)$$

where $\frac{1}{2}\|u(t, Y_t^u)\|^2$ and $-\log \tilde{h}_T(Y_T^u)$ represent the running cost and terminal cost, respectively. The expectation is taken with respect to the solution $\{Y_t^u\}_{t \in [0,T]}$ of the controlled SDE

$$dY_t^u = \left(\bar{f}_\theta(t, Y_t^u) + \bar{g}(t)u(t, Y_t^u)\right) dt + \bar{g}(t)d\bar{W}_t, \quad (20)$$

with initial condition $Y_0^u \sim P_{\text{noise}}$.

Assuming the existence of an optimal control, we define the value function as

$$V(t, x) = \inf_u \mathbb{E}_x\left[\int_t^T \frac{1}{2}\|u(s, Y_s^u)\|^2\, ds - \log \tilde{h}_T(Y_T^u)\right], \quad (21)$$

where $\mathbb{E}_x$ denotes the expectation conditioned on $Y_t^u = x$. It is well known that the value function $V$ satisfies the Hamilton–Jacobi–Bellman (HJB) equation, a nonlinear partial differential equation associated with this control problem. Moreover, the optimal control policy can be expressed explicitly in terms of the value function $V$.

**Proposition 5.1.** *Consider the stochastic optimal control problem with cost functional (19) and controlled SDE (20). Let $V$ denote the value function defined in (21), and consider the exponential transformation $\tilde{h}(t, x) = \exp(-V(t, x))$. Then, the function $\tilde{h}$ solves the backward Kolmogorov equation (8) with terminal condition $\tilde{h}(T, x) = \tilde{h}_T(x)$. Moreover, the optimal control is given by*

$$u^*(t, x) = \bar{g}(t)\nabla_x \log \tilde{h}(t, x), \quad (22)$$

*and the resulting closed-loop dynamics satisfy*

$$dY_t = \left(\bar{f}_\theta(t, Y_t) + \bar{g}(t)u^*(t, Y_t)\right) dt + \bar{g}(t)\, d\bar{W}_t. \quad (23)$$

*This closed-loop SDE coincides with the guided reverse SDE* (13) *constructed via approximate Doob's $h$-transform using the function $\tilde{h}$.*

This formulation enables the use of established techniques from stochastic optimal control to compute guidance functions for guided diffusion models. One useful approach is the iterative Linear Quadratic Gaussian (iLQG) method (Todorov & Li, 2005), which constructs local quadratic approximations of the dynamics and cost around a nominal trajectory and iteratively refines the control policy. This procedure can be interpreted as a Newton-type method for optimal control problems. Another useful approach is path integral control (Kappen, 2005; Theodorou et al., 2010), which reformulates the stochastic optimal control problem in terms of a path integral and estimates the optimal control using Monte Carlo sampling.

The path integral control is introduced as follows.

**Theorem 5.2.** *Consider the stochastic optimal control problem with the cost functional* (19) *and the controlled SDE* (20). *Then, the optimal control $u^*$ minimizing the cost functional* (19) *is given by*

$$u^*(t, x) = u_b(t, x) +$$

$$\lim_{\Delta t \to 0} \frac{1}{\Delta t} \frac{\mathbb{E}\left[\exp\left(-S^u(t)\right) \int_t^{t+\Delta t} d\bar{W}_s \mid Y_t = x\right]}{\mathbb{E}\left[\exp\left(-S^u(t)\right) \mid Y_t = x\right]}, \quad (24)$$

*where $u_b$ is an arbitrary baseline control function, and $S^{u_b}(t)$ is the cost functional defined by*

$$S^{u_b}(t) =$$
$$\int_t^T \frac{1}{2}\|u_b(s, Y_s)\|^2 ds + \int_t^T u_b(s, Y_s)^T d\bar{B}_s + V_T(Y_T), \quad (25)$$

*where the conditional expectations are taken with respect to trajectories generated by* (20) *under $u_b$, $S^{u_b}(t)$ is defined in* (25), *and $V_T(x) := -\log \tilde{h}_T(x)$.*

The proof of Theorem 5.2 is provided in Appendix G. Theorem 5.2 shows that the optimal control (24) can be computed by evaluating expectations over trajectories of the controlled SDE (20) under an arbitrary baseline control $u_b$, including the uncontrolled case $u_b \equiv 0$. In this formulation, the terminal cost $V_T(x) = -\log \tilde{h}_T(x)$ is given by the log transform of a smooth approximation of the indicator function $\mathbf{1}_S(x)$. This representation leads to a practical and computationally efficient procedure for computing the guidance term in the guided reverse SDE (7), as it relies only on Monte Carlo sampling and does not require solving the HJB equation or introducing additional learning procedures.

From the viewpoint developed here, we obtain a principled framework for guidance design. In this framework, the terminal cost function plays a central role in determining

the resulting guidance. Furthermore, if the terminal cost is designed such that the induced guidance approximates the ideal Doob guidance, then Theorem 4.4 guarantees a high success probability. Additionally, the fact that the terminal cost design is crucial implies that we can devise the design to include the information, such as the language instructions, as discussed in the following section.

The viewpoint developed in this section also provides insight into simple guidance mechanisms based on a time-independent potential function $J(x)$, leading to guidance of the form $u(t, x) = -\bar{g}(t)\nabla_x J(x)$. This idea is introduced in (Sohl-Dickstein et al., 2015) for discrete-time diffusion models. The discussion above implies that such a function $J(x)$ can be interpreted as specifying an approximate terminal cost in the associated stochastic optimal control formulation. Moreover, if the induced guidance $-\bar{g}(t)\nabla_x J(x)$ is sufficiently close to the ideal Doob guidance $\bar{g}(t)\nabla_x \log h(t, x)$ in the sense quantified by (16), then Theorem 4.4 provides theoretical support for expecting the success probability to remain high.

## 6. Application to Planning

We present an outline for adapting the guidance method based on Doob's $h$-transform to planning problems, following the diffusion-based planning framework of (Janner et al., 2022) without considering rewards.

We consider the planning problem of determining a trajectory of the controlled discrete-time dynamical system given by

$$s_{k+1} = F(s_k, a_k) \text{ for } k = 0, \ldots, K-1 \quad (26)$$

where $s_k \in \mathbb{R}^{n_s}$ denotes the state variable, $a_k \in \mathbb{R}^{n_a}$ denotes the action variable, and $F : \mathbb{R}^{n_s} \times \mathbb{R}^{n_a} \to \mathbb{R}^{n_s}$ denotes the state transition function. Here, the planning index $k$ should be distinguished from the time variable $t$ in the reverse-time SDE. We assume that the initial state $s_0$ is given. Note that even when the function $F$ is unknown, diffusion models can learn the trajectory distribution from trajectory data.

The diffusion model (5) is trained to generate samples of trajectories of the form $\tau = (s_0, a_0, s_1, a_1, \ldots, s_K) \in \mathbb{R}^{(n_s+n_a)K+n_s}$, so that it learns the trajectory distribution of the system from trajectory data on $\mathbb{R}^{(n_s+n_a)K+n_s}$. Given the trained score function in (5), a sample trajectory $\tau$ is generated as $\tau = Y_T$, where $\{Y_t\}_{t \in [0,T]}$ is the solution to the corresponding reverse-time SDE.

Then, given the trained diffusion model, the guidance design based on Doob's $h$-transform can be applied to the reverse-time SDE to generate a trajectory that satisfies a prescribed condition, such as reaching a target state or satisfying a constraint. The general procedure for adapting the guidance design based on Doob's $h$-transform to planning

---

**Algorithm 1** Doob-Guided Diffusion Sampling

---

**Require:** Score network $s_\theta$, terminal function $\tilde{h}_T$, diffusion steps $N$
**Require:** Monte Carlo samples $M$, step size $\Delta t = T/N$
1: Sample $Y_0 \sim \mathcal{N}(0, I)$      // Initialize from noise
2: **for** $k = 0$ to $N - 1$ **do**
3:     $t_k \leftarrow k \cdot \Delta t$
4:     $s_k \leftarrow s_\theta(T - t_k, Y_k)$      // Compute score
5:     $u_k \leftarrow \bar{g}(t_k)^2 \nabla \log \tilde{h}(t_k, Y_k)$    // Guidance
6:     $\epsilon \sim \mathcal{N}(0, I)$
7:     $Y_{k+1} \leftarrow Y_k + (\bar{f}_\theta(t_k, Y_k) + u_k)\Delta t + \bar{g}(t_k)\sqrt{\Delta t} \cdot \epsilon$
8: **end for**
9: **return** $Y_N$      // Generated trajectory

---

problems is as follows. First, we determine the target set $S \subset \mathbb{R}^{(n_s + n_a)K + n_s}$ that characterizes the desired property of the trajectory, such as reaching a target state or satisfying a constraint. Then, we determine the function $\tilde{h}_T$ in (19) as a smooth approximation of the indicator function $\mathbf{1}_S$ of the target set $S$. A typical example of the smooth approximation function $\tilde{h}_T$ is given by $\tilde{h}_T(\tau) = \exp(-\lambda d(\tau, S)^2)$, where $\lambda > 0$ is a tunable parameter and $d(\tau, S)$ is a distance function that measures the distance between the trajectory $\tau$ and the target set $S$, for example, the distance between the final state of the trajectory and the target state. Lastly, we compute the guidance function $u^*(t, x)$ using methods such as iLQG or path integral control, as presented in Section 5, and apply the resulting guidance to the controlled SDE (20), which yields the guided reverse-time SDE, to generate a trajectory satisfying the prescribed condition.

# 7. Numerical Experiments

We evaluate our path integral control (PIC) guidance and iLQG-based guidance on robotic planning tasks, demonstrating sample efficiency, safety, and generalization across environments of increasing complexity and dimensionality. We consider three evaluation axes: (i) constraint satisfaction and success rate on 2D navigation benchmarks, (ii) out-of-distribution generalization to novel obstacle configurations, and (iii) zero-shot transfer to higher-dimensional robotic manipulation. See Appendix L for additional experimental details.

## 7.1. Experimental Setup

We consider 2D navigation with discrete-time point-mass dynamics $s_{k+1} = s_k + v_k \Delta t$, where $k$ indexes the planning timestep, $s_k \in \mathbb{R}^2$ denotes the position and $v_k \in \mathbb{R}^2$ is the control velocity. The diffusion model is trained on 10,000 expert trajectories collected from the *Simple* scenario (1 obstacle); evaluation is performed on increasingly complex out-of-distribution environments, including *Medium* (3 ob-

stacles) and *HardTrap* (4 obstacles with narrow passages) (see Figure 1). All experiments are conducted using five random seeds with 100 trials per seed, and results are reported as the mean with 95% confidence intervals. We further evaluate zero-shot transfer to a 7-DOF Franka Panda robot in MuJoCo (Todorov et al., 2012), where the 2D-trained diffusion model plans in task space and trajectories are mapped to joint configurations via inverse kinematics (IK).

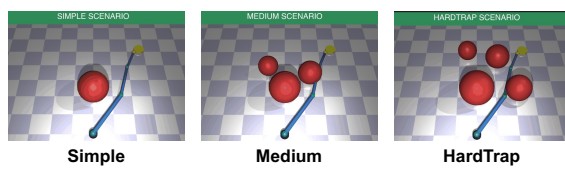

*Figure 1.* Illustration of 2D navigation scenarios.

## 7.2. Results and Analysis

**Comprehensive Benchmark.** We compare the proposed PIC and iLQG guidance against two baseline approaches: (i) *Reflected Diffusion* (Fishman et al., 2023a), a learning-based approach that enforces constraints through reflection during the diffusion process; and (ii) *NoGuidance*, which samples trajectories directly from the diffusion model without any explicit constraint handling. As shown in Table 1, iLQG achieves perfect feasibility across all scenarios, providing an oracle reference. The proposed PIC guidance closely matches this upper-bound performance across all environments, including the out-of-distribution HardTrap scenarios, despite operating purely as an inference-time guidance mechanism on top of a learned diffusion model. In contrast, unguided diffusion fails to produce feasible trajectories in all constrained settings, highlighting the necessity of explicit constraint-aware guidance. We further benchmark PIC against additional baselines including MPPI, Gradient Guidance, Inference Guidance, and Diffusion Inpainting on a set of challenging topological scenarios; see Appendix L.6 for details.

| Method | Simple | Medium | HardTrap |
|---|---|---|---|
| PIC (Ours) | $89.3 \pm 3.8$ | $90.7 \pm 1.9$ | $88.0 \pm 3.3$ |
| iLQG (Ours) | $100.0 \pm 0.0$ | $100.0 \pm 0.0$ | $100.0 \pm 0.0$ |
| Reflected Diffusion | $64.0 \pm 6.5$ | $69.3 \pm 3.8$ | $68.0 \pm 3.3$ |
| NoGuidance | $0.0 \pm 0.0$ | $0.0 \pm 0.0$ | $0.0 \pm 0.0$ |

*Table 1.* Success rates (%) with 95% confidence intervals on 2D navigation tasks (100 trials, 5 seeds, $n = 64$ MC samples).

**Zero-Shot Transfer.** We evaluate zero-shot transfer of the 2D-trained model to a 7-DOF Franka Panda robot across three scenarios (Wall, Corridor, Dense Forest) in MuJoCo. The 2D diffusion model plans in task space; trajectories are mapped to joint configurations via inverse kinematics. As shown in Table 2, baseline methods achieve perfect safety but zero success, as 2D planning followed by post-hoc IK

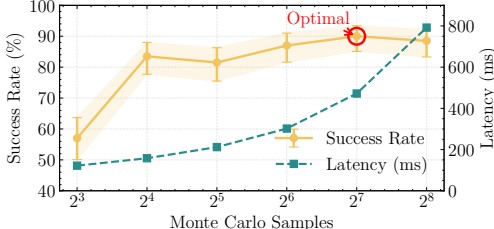

*Figure 2.* Effect of Monte Carlo samples $n$ on PIC performance (HardTrap). Success improves from $57.0\%$ at $n = 8$ to $90.0\%$ at $n = 128$, with safety above $89\%$ throughout.

fails to account for full-body geometry in 3D environments. In contrast, PIC maintains safety while successfully reaching the goal across all scenarios, demonstrating that training-free guidance generalizes zero-shot to higher-dimensional systems without any task-specific retraining.

| Method | Wall | Corridor | Dense Forest |
|---|---|---|---|
| 2D MPPI + IK | 100 / 0 | 100 / 0 | 100 / 0 |
| Diffusion Inpaint. + IK | 100 / 0 | 100 / 0 | 100 / 0 |
| PIC (Ours) | 100 / 60 | 100 / 40 | 100 / 100 |

*Table 2.* Safety / Success (%) for zero-shot transfer to a 7-DOF Franka Panda in MuJoCo. All methods achieve 100% safety; only PIC reaches the goal.

**Ablation on Path Integral Approximation.** We analyze the sensitivity of the proposed method to the number of Monte Carlo samples $n$ used in the path integral computation on the HardTrap scenario (see Figure 2). As $n$ increases from 8 to 128, success rate improves steadily from $57.0\%$ to $90.0\%$, indicating more accurate approximation of the optimal guidance. Notably, safety remains above $89\%$ across all budgets, demonstrating that PIC's safety guarantees are robust regardless of the MC budget. Performance peaks at $n = 128$, achieving $90.0\%$ success with a latency of $472$ ms, while $n = 64$ offers a favorable trade-off at $87.0\%$ success and $302$ ms. Based on this trade-off between performance and computational cost, we use $n = 64$ as the default budget in all subsequent experiments.

**Out-of-Distribution Generalization.** We evaluate zero-shot generalization of PIC to environments with increasing obstacle density, ranging from 5 to 50 randomly placed obstacles across 100 procedurally generated configurations (see Figure 3). PIC achieves an overall success rate of $91.4\%$ $[88.6, 93.6]$ while maintaining $100\%$ safety across all obstacle densities, demonstrating that the training-free guidance mechanism generalizes robustly to environments far beyond the training distribution of a single obstacle. Notably, safety remains perfect even at 50 obstacles, suggesting that PIC's constraint satisfaction guarantees hold under significant distributional shift.

**Language-Conditioned Planning.** We evaluate language-

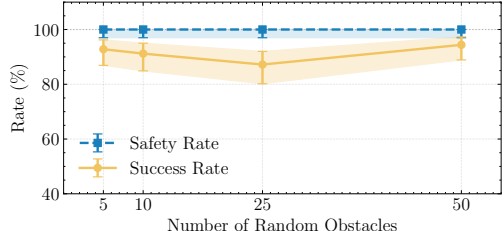

*Figure 3.* Out-of-distribution generalization.

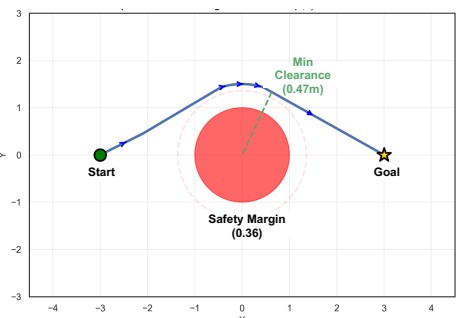

*Figure 4.* Language-guided planning to avoid the red region.

conditioned cost specification across the Simple, Medium, and HardTrap scenarios, where natural language commands (e.g., *"navigate safely to the goal while avoiding all obstacles"*) are converted into soft cost functions $\tilde{h}_T$ via GPT-4; here the LLM acts as a cost compiler rather than a planner, mapping language instructions into differentiable terminal cost functions (see Figure 4). PIC achieves $88.5\%$ success rate on language-conditioned tasks compared to $42.3\%$ for cost-function replacement baselines and $0\%$ for unguided diffusion. This validates that the path integral framework can accommodate language-derived specifications while enforcing feasibility through the stochastic optimal control formulation. See Appendix L.7 for details on the language-to-cost compilation pipeline and GPT-4 prompts.

## 8. Conclusion

This study presented a theoretical framework for guided diffusion models grounded in Doob's $h$-transform. The framework characterizes Doob's $h$-transform as an ideal guidance mechanism that guarantees satisfaction of prescribed conditions, and formalizes practical guidance design as an approximation to this ideal. Within this framework, we derived explicit bounds on the success probability of guided diffusion. We further showed that guidance design admits a stochastic optimal control formulation, enabling efficient and training-free guidance for robotic planning. We validated the proposed framework on robotic planning tasks, demonstrating reliability, generalization, and efficiency, including language-conditioned cost specifications.

## Acknowledgements

The authors would like to thank the reviewers and area chairs for their valuable comments.

This work was supported in part by JSPS KAKENHI (grant no. 23K03902), by JST ACT-X (grant no. JPMJAX210L), JST PRESTO (grant no. JPMJPR2516), and by NSF CAREER (grant no. 2442948).

## Impact Statement

This study focuses on the theoretical analysis of guided diffusion models. Recent advances in generative models have enabled their application to autonomous systems, including robotics. However, theoretical understanding of their behavior remains limited. In safety-critical settings, such as autonomous systems, reliability is an important consideration. This study seeks to contribute to the theoretical understanding of guided diffusion models by developing an analytical framework. We hope that such insights may eventually support more reliable use of diffusion-based methods in autonomous systems.

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

# A. Notation

We show the notation used throughout the paper.

We denote the set of real numbers by $\mathbb{R}$ and the set of non-negative real numbers by $\mathbb{R}_{\geq 0}$. The $n$-dimensional Euclidean space is denoted by $\mathbb{R}^n$ and the norm on $x \in \mathbb{R}^n$ is denoted by $\|x\| := \sum_{i=1}^{n} |x_i|^2$. The notation $\mathbb{R}^{n \times m}$ denotes the set of matrices with $n$ rows and $m$ columns. For the vector $x \in \mathbb{R}^n$ and the matrix $A \in \mathbb{R}^{n \times m}$, the transposes of $x$ and $A$ are denoted by $x^T$ and $A^T$, respectively.

We denote random variables by uppercase letters such as $X$ and their realizations by lowercase letters such as $x$. We also denote probability distributions by uppercase letters such as $P$ and their density functions by lowercase letters such as $p$. When the random variable $X$ follows the probability distribution $P$, we write $X \sim P$. To represent the probability distribution (law) of a random variable $X$, we sometimes use the notation $\mathcal{L}(X)$. Expectations are written as $\mathbb{E}_P[\cdot]$ or $\mathbb{E}_X[\cdot]$ when the dependence is explicit. We usually use $W_t$ or $B_t$ to denote a Brownian motion throughout this paper. The the indicator function of the target set $S$ is defined as

$$\mathbf{1}_S(x) = \begin{cases} 1, & x \in S, \\ 0, & x \notin S. \end{cases} \tag{27}$$

# B. Related Works

## B.1. Optimization and constraints in diffusion models

*Diffusion and stochastic optimal control.* Recent work has begun to connect diffusion with stochastic optimal control methods. Li & Pereira (2024) recast diffusion-based inverse problems as an optimal control problem, and Zhang & Chen (2022); Behjoo & Chertkov (2025) consider reverse processes in diffusion models as a stochastic optimal control problem. Li & Chen (2025) explains the similarities in the update structure of reverse diffusion sampling and Model Predictive Path Integral Control under a Gibbs-measure/energy-optimization viewpoint. For guided diffusion, importance-sampling and optimal control have been used to obtain training-free guidance to generate or select among multiple candidate transitions during sampling (Kim et al., 2025) and for symbolic music generation (Huang et al., 2024). Conversely, Huang & Liu (2024) explores the application of diffusion models to improve MPPI-style planning. In comparison, our approach starts from Doob's $h$-transform characterization of reverse-time diffusions and leverages a path-integral-control formulation to compute the corresponding guidance.

*Theoretical studies on diffusion models.* Existing theoretical analyses of diffusion models have primarily focused on convergence and sampling properties for *unguided* dynamics, often quantified via total variation or Wasserstein-type metrics, among others (Chen et al., 2023; Tang & Zhao, 2024; De Bortoli et al., 2021); see also Tang & Zhao (2025) for an overview. These results, however, do not directly carry over to guided diffusion, especially the evaluation of the success probability of guided diffusion.

*Diffusion with constraints.* For applications requiring generated samples to satisfy constraints (e.g., safety, feasibility, or domain validity), recent work has incorporated constraints directly into diffusion dynamics and/or inference. Reflected diffusion models incorporate data constraints by considering the forward and backward process of reflected stochastic differential equations that evolve on the support of data (Lou & Ermon, 2023). Barrier diffusion models introduce additional potential functions to penalize and confine diffusion paths when approaching invalid regions. Efficient techniques based on Metropolis sampling are developed for reflected or barrier diffusions (diffusion models with reflected Brownian motion or logarithmic barrier metric), which also allows these techniques to be applied to non-convex data constraints (Fishman et al., 2023b). These techniques are also extended to the Riemannian manifold for the construction of Riemannian diffusion models (Fishman et al., 2023a). Projected diffusion projects data points outside of the valid regions back to the feasible set. Constrained diffusion problem has also been formulated into a constrained distributional optimization problem that optimizes a distribution similar to the data distribution while satisfying data constraints (Khalafi et al., 2024). Mirror diffusion constructs dual-space diffusion processes for tractable conditional scores and simulation-free training (Liu et al., 2023). Those approaches have not adopted path integral control.

## B.2. Diffusion for decision making

*Diffusion for imitation learning.* Score-based diffusion models provide a flexible way to parameterize stochastic policies and trajectory generators for decision making. In imitation learning, action-diffusion policies learn observation-conditioned

action distributions (Chi et al., 2025), with work on improved guidance and sampling (Pearce et al., 2023), robustness via constrained-context denoising (Saxena et al., 2023), and goal-conditioned score-based diffusion policies that can be sampled in a few steps (Reuss et al., 2023). For long-horizon manipulation, diffusion policies have been combined with temporal abstraction and hierarchical structure, e.g., chaining keypose prediction with trajectory diffusion (Xian et al., 2023), kinematics-aware hierarchical diffusion policies (Ma et al., 2024), and stateful diffusion policies that reduce action ambiguity over time (Liu et al., 2024). Diffusion can also be used to augment imitation objectives or datasets, such as combining BC with diffusion-model losses (Chen et al., 2024) or synthesizing corrective data in a DAgger-like process (Zhang et al., 2024b).

*Diffusion for reinforcement learning.* Diffusion Q-learning injects Q-function guidance into diffusion training to trade off return maximization and behavior regularization (Wang et al., 2023), and later work accelerates training/sampling and connects diffusion policies to maximum-likelihood RL objectives (Kang et al., 2023). Energy-guided diffusion offers a principled route to incorporate reward or constraint signals at sampling time; contrastive energy prediction learns exact intermediate guidance and has been applied to offline RL (Lu et al., 2023). Data-centric pipelines such as diffusion-based trajectory stitching can further improve downstream RL by synthesizing connecting transitions (Li et al., 2024). Diffusion policies have also been optimized in online settings, including policy-gradient fine-tuning of pretrained diffusion policies (Ren et al., 2025) and diffusion-based actor-critic variants that optimize a Q-weighted objective or incorporate entropy regularization (Ding et al., 2024; Wang et al., 2024b). In adversarial imitation learning, diffusion models have been used to construct discriminators (Lai et al., 2024) and to model the distribution of system trajectories for distinguishing expert demonstrations from those generated by a learned policy (Wang et al., 2024a).

*Diffusion for planning.* Diffusion models have also been used for planning by modeling trajectory distributions and sampling plans conditioned on start/goal, costs, or returns. Diffuser shows trajectory diffusion as a flexible offline planner (Janner et al., 2022), and recent work distills/accelerates diffusion planners for high-frequency control (Lu et al., 2025). Diffusion priors can be combined with explicit costs or optimization to enforce collision and feasibility constraints, including ensemble-of-costs guidance (Saha et al., 2024), potential-based diffusion motion planning (Luo et al., 2024), and diffusion-generated seeds for fast motion optimization (Huang et al., 2025). Diffusion models have also been integrated into MPC by learning both dynamics and action proposal distributions (Zhou et al., 2025), and constrained multi-robot planning via hybrid classical-learning or projection-based sampling schemes (Shaoul et al., 2025; Liang et al., 2025). In comparison, we develop a guidance framework for continuous-time diffusion models based on Doob's $h$-transform and path integral control, with theoretical guarantees for approximate Doob guidance.

### B.3. Language guided diffusion

Language-guided diffusion has been most extensively developed in text-to-image generation (see (Cao et al., 2025) for a review). Recent literature has studied how to take language guidance for other forms of data, such as samples from time series and dynamical systems. Some techniques design and train networks with embedded text encoders to directly take language as input (Vuong et al., 2023; Chen & Li, 2023). In robotics and planning, language can be incorporated into diffusion models by conditioning on language-derived latent variables, cost functions, or constraint sets. Li et al. (2025) incorporates language input as the condition variables in conditional diffusion models (Yu et al., 2023), and Zhang et al. (2024a) couples a language-conditioned diffusion planner with a lower-level controller for real-time control. Janner et al. (2022); Zhong et al. (2023); Ziyuan et al. (2023) incorporates language input as the guidance function to guide the data generation process. Naively using cost functions generated from language models as the guidance functions can yield suboptimal or unsafe performance, as those functions may not account for the system dynamics and feasibility.

## C. Brief Review of Score-based Diffusion Models

This section briefly provides an overview of score-based continuous-time diffusion models as a prerequisite for the main results of this study. The diffusion models have been develop in Sohl-Dickstein et al. (2015); Ho et al. (2020); Song et al. (2021). We here focus on the continuous-time score-based diffusion models (Song et al., 2021), which is the basis of this study.

As developed in Sohl-Dickstein et al. (2015); Ho et al. (2020); Song et al. (2021), generative modeling using the diffusion models consists of two processes: the forward diffusion process and the reverse diffusion process, which are modeled by the SDEs in the continuous-time setting (Song et al., 2021). Given the data distribution $P_{\text{data}}$, the forward process corresponds

to gradually corrupting the data by adding noise, modeled by the SDE

$$dX_t = f(t, X_t)dt + \sigma(t)dB_t, \quad X_0 \sim P_{\text{data}} \tag{28}$$

where $\sigma : [0, T] \to \mathbb{R}$ is a scalar function and $B_t$ is the $n$-dimensional standard Brownian motion. Then, the solution of (28) determines the probability distribution $\mathcal{L}(X_t)$ for each $t \in [0, T]$ starting from $\mathcal{L}_{\text{data}}$ to a simple distribution typically towards a Gaussian distribution. We denote the probability density function of the distribution $\mathcal{L}(X_t)$ by $p(t, x)$.

The reverse process corresponds to the process of generating new data samples by reversing the forward diffusion process. The reverse process is given by the reverse-time SDE:

$$dY_t = \bar{f}(t, Y_t)dt + \bar{\sigma}(t)d\bar{B}_t, \tag{29}$$

where $\bar{B}_t$ is the $n$-dimensional standard Brownian motion, and the drift term $\bar{f} : [0, T] \times \mathbb{R}^n \to \mathbb{R}^n$ and the diffusion term $\bar{\sigma} : [0, T] \to \mathbb{R}$ are given by

$$\bar{f}(t, x) = -f(T - t, x) + \sigma(T - t)^2 \nabla \log p(T - t, x), \quad \bar{\sigma}(t) = \sigma(T - t). \tag{30}$$

where $p(t, x)$ is the probability density function of the solution to the forward SDE (28) at time $t$. Note that for the reverse-time SDE (29), this study assumes the time variable $t$ runs from $0$ to $T$, although literature assumes the time variable runs from $T$ to $0$. In theory, the reverse diffusion process starts from the distribution $\mathcal{L}(X_T)$, which is a marginal distribution of the solution to the forward process (28) at time $T$, to ensure that

$$\mathcal{L}(Y_t) = \mathcal{L}(X_{T-t}), \tag{31}$$

for all $t \in [0, T]$. In practice, the distribution $\mathcal{L}(X_T)$ is approximated by a simple distribution that approximates $\mathcal{L}(X_T)$, such as a Gaussian distribution.

Since the score function $\nabla \log p(t, x)$ is intractable to compute, the reverse process is approximated by replacing the score function with a score network $s_\theta : [0, T] \times \mathbb{R}^n \to \mathbb{R}^n$, resulting in the approximate reverse SDE:

$$dY_t = \bar{f}_\theta(t, Y_t)dt + \bar{\sigma}(t)d\bar{B}_t, \tag{32}$$

where the drift term $\bar{f}_\theta : [0, T] \times \mathbb{R}^n \to \mathbb{R}^n$ is given by

$$\bar{f}_\theta(t, x) = -f(T - t, x) + \sigma(T - t)^2 s_\theta(T - t, x). \tag{33}$$

The score network $s_\theta$ is trained by minimizing the score matching loss function

$$L(\theta) = \mathbb{E}_{t \sim \mathcal{U}(0,T)} \left[ \sigma(t)^2 \mathbb{E}_{X_0, X_t} \left[ \|s_\theta(t, X_t) - \nabla \log p(X_t|X_0)\|^2 \right] \right], \tag{34}$$

where $\mathcal{U}(0, T)$ is the uniform distribution on the interval $(0, T)$, $p(X_t|X_0)$ is the transition probability density of the forward SDE (1) given the initial value $X_0$. This ensures that $\nabla \log p(t, x) \approx s_\theta(t, x)$ for $t \in (0, T)$ and $x \in \mathbb{R}^n$.

The sampling via the diffusion model is performed by simulating the score-based reverse SDE (32) starting from the simple distribution approximating $\mathcal{L}(X_T)$. The solution $Y_T$ is used as a new sample from the data distribution $P_{\text{data}}$.

There are various choices for the drift term $f$ and the diffusion term $\sigma$ in the forward SDE (28). The typical choices are the Variance Exploding (VE) SDE and the Variance Preserving (VP) SDE (Song et al., 2021).

**Variance Exploding (VE) SDE** The variance exploding SDE is defined by

$$f(t, x) = 0, \quad g(t) = \sqrt{\frac{d\sigma^2(t)}{dt}}, \quad \sigma(t) = \sigma_{\min} \left( \frac{\sigma_{\max}}{\sigma_{\min}} \right)^{t/T} \sqrt{\frac{2}{T} \log \left( \frac{\sigma_{\max}}{\sigma_{\min}} \right)}, \tag{35}$$

where $\sigma_{\min} > 0$ and $\sigma_{\max} > \sigma_{\min}$ are some constants.

**Variance Preserving (VP) SDE** The variance preserving SDE is defined by

$$f(t, x) = -\frac{1}{2}\beta(t)x, \quad \sigma(t) = \sqrt{\beta(t)}, \tag{36}$$

where $\beta : [0, T] \to \mathbb{R}$ is usually chosen as

$$\beta(t) = \beta_{\min} + t(\beta_{\max} - \beta_{\min}), \tag{37}$$

with some constants $0 < \beta_{\min} < \beta_{\max}$.

## D. Brief Review of Doob's $h$-transform

This section briefly reviews the Doob's $h$-transform. Detailed exposition can be found in (Doob, 1984; Rogers & Williams, 2000; Särkkä & Solin, 2019).

The Doob's $h$-transform is a theory to construct a Markov process conditioned on a specific event, particularly reaching a certain region at a given time. The theory can be applied to the general Markov processes, but we focus on the case of the SDEs yielding a Markov process.

Throughout this section for Doob's $h$-transform, we consider the SDE, not limited to the score-based diffusion models, given by

$$dZ_t = b(t, Z_t)dt + \sigma(t, Z_t)dW_t \tag{38}$$

where $b : [0, T] \times \mathbb{R}^n \to \mathbb{R}^n$ is the drift term, $\sigma : [0, T] \times \mathbb{R}^n \to \mathbb{R}^{n \times m}$ is the diffusion term, and $W_t$ is the $m$-dimensional standard Brownian motion. The SDE (38) yields the infinitesimal generator of the Markov process defined by

$$L_t \phi(x) = \nabla \phi(x)^T b(t, x) + \frac{1}{2} \mathrm{tr} \left( \sigma(t, x) \sigma(t, x)^T \nabla^2 \phi(x) \right) \tag{39}$$

for a smooth function $\phi : \mathbb{R}^n \to \mathbb{R}$.

Then, we assume that the SDE (38) defines the transition probability density function $p(t', y|t, x)$ for $t, t' \in [0, T]$ and $x, y \in \mathbb{R}^n$. Then, Doob's $h$-transform considers the transformation of the transition probability density function $p(t', y|t, x)$ given by

$$p^h(t', y|t, x) = \frac{h(t', y)}{h(t, x)} p(t', y|t, x) \tag{40}$$

where $h : [0, T] \times \mathbb{R}^n \to \mathbb{R}_{>0}$ is a positive function satisfying the backward Kolmogorov equation

$$\frac{\partial h}{\partial t}(t, x) + L_t h(t, x) = 0, \quad h(T, x) = h_T(x), \tag{41}$$

where $h_T : \mathbb{R}^n \to \mathbb{R}_{>0}$ is a positive function. The new transition probability density function $p^h(t', y|t, x)$ defines a new Markov process based on the original Markov process defined by (38), referred to as the Doob's $h$-transform. The new Markov process is given by the SDE

$$dZ_t^h = \left( b(t, Z_t^h) + \sigma(t, Z_t^h)\sigma(t, Z_t^h)^T \nabla \log h(t, Z_t^h) \right) dt + \sigma(t, X_t^h)dW_t, \tag{42}$$

where $h(t, x)$ is the solution to the backward Kolmogorov equation (41). Because of the construction, the solution $Z_t^h$ has the transition probability density function $p^h(t', y|t, x)$.

The Doob's $h$-transform is employed to construct a Markov process conditioned on reaching a prescribed set at a given time, particularly at the terminal time. When we consider to condition the SDE (38) on the terminal value $Z_T = z$ for some $z \in \mathbb{R}^n$, the function $h(t, x)$ is given by

$$h(t, x) = p(T, z|t, x) \tag{43}$$

where $p(T, z|t, x)$ is the transition probability density function of the SDE (38) given $z \in \mathbb{R}^n$. Then, constructing the new transition density function as

$$p^h(t', y|t, x) = \frac{p(T, z|t', y)}{p(T, z|t, x)} p(t', y|t, x), \tag{44}$$

leading to

$$\lim_{t' \to T} p^h(t', y|t, x) = \delta_z(y), \tag{45}$$

where $\delta_z$ is the Dirac delta function centered at $z$ under appropriate conditions.

## E. Brief Review of Stochastic Optimal Control

We briefly review a standard formulation of stochastic optimal control, which motivates the path integral control framework employed in the main paper.

We consider a controlled SDE of the form

$$dZ_t = (b(t, Z_t) + \sigma(t, Z_t)u_t)\, dt + \sigma(t, Z_t)dW_t, \quad 0 \le t \le T, \tag{46}$$

where $b$ and $\sigma$ denote the drift and diffusion coefficients, respectively, and $u_t$ represents the control input process.

Throughout this appendix, we restrict attention to controlled SDEs of the form (46). This specific structure allows us to apply Girsanov's theorem to characterize the induced change of measure, and forms the basis of the path integral control formulation.

The goal of the control problem is to design a control policy that optimizes a prescribed performance objective. A standard formulation is given by the stochastic optimal control problem of minimizing the cost functional

$$J(t, z, u) = \mathbb{E}_z \left[ \int_t^T \ell(Z_s, u_s)\, ds + \varphi(Z_T) \right], \tag{47}$$

subject to the controlled SDE (46), where $\ell : \mathbb{R}^n \times \mathbb{R}^m \to \mathbb{R}$ denotes the running cost, $\varphi : \mathbb{R}^n \to \mathbb{R}$ denotes the terminal cost, and $\mathbb{E}_z$ denotes expectation with respect to the solution of (46) starting from $Z_t = z$.

The associated value function is defined as the infimum of the expected cost over all admissible control processes:

$$V(t, z) = \inf_u \mathbb{E}_z \left[ \int_t^T \ell(Z_s, u_s)\, ds + \varphi(Z_T) \right]. \tag{48}$$

It is well known that the value function satisfies a nonlinear partial differential equation known as the Hamilton–Jacobi–Bellman (HJB) equation. In this paper, the HJB equation plays a key role in establishing the control-theoretic characterization of Doob's $h$-transform. The following result can be found in (Fleming & Soner, 2006).

**Theorem E.1** (Hamilton–Jacobi–Bellman equation)**.** *Under standard regularity assumptions, the value function $V(t, x)$ associated with the stochastic optimal control problem defined by the cost functional* (47) *and the controlled SDE* (46) *satisfies the HJB equation*

$$\partial_t V(t, x) + \inf_{u \in \mathbb{R}^m} [L_t^u V(t, x) + \ell(x, u)] = 0, \quad V(T, x) = \varphi(x), \tag{49}$$

*where $\varphi : \mathbb{R}^n \to \mathbb{R}$ is the terminal cost.*

The HJB equation can be derived from the dynamic programming principle. A detailed proof of Theorem E.1 can be found in standard references, such as (Fleming & Soner, 2006).

*Remark* E.2. Throughout this appendix, we focus on classical solutions of the HJB equation and do not consider the more general notion of viscosity solutions.

When the running cost is chosen in the quadratic form

$$\ell(x, u) = \frac{1}{2}\|u\|^2, \tag{50}$$

we obtain the following corollary of Theorem E.1.

**Corollary E.3.** *Under standard regularity assumptions, the value function $V(t, x)$ associated with the stochastic optimal control problem defined by the cost functional* (47) *and the controlled SDE* (46) *satisfies the nonlinear PDE*

$$\partial_t V(t, x) + \nabla_x V(t, x)^\top b(t, x) + \frac{1}{2}\mathrm{Tr}\left(\sigma(t, x)\sigma(t, x)^\top \nabla_x^2 V(t, x)\right) - \frac{1}{2}\left\|\sigma(t, x)^\top \nabla_x V(t, x)\right\|^2 = 0, \tag{51}$$

*with terminal condition $V(T, x) = \varphi(x)$. Moreover, the optimal control is given in feedback form by*

$$u^*(t, x) = -\sigma(t, x)^\top \nabla_x V(t, x). \tag{52}$$

The following proof is standard, but we include it for completeness.

*Proof.* It suffices to evaluate the pointwise minimization problem

$$\inf_{u\in\mathbb{R}^m}\left[L_t^u V(t,x)+\frac{1}{2}\|u\|^2\right]. \tag{53}$$

By definition of the controlled generator, we have

$$L_t^u V(t,x)+\frac{1}{2}\|u\|^2=\nabla_x V(t,x)^\top b(t,x)+\frac{1}{2}\text{Tr}\big(\sigma(t,x)\sigma(t,x)^\top\nabla_x^2 V(t,x)\big)+\nabla_x V(t,x)^\top\sigma(t,x)u+\frac{1}{2}u^\top u. \tag{54}$$

Completing the square with respect to $u$, we obtain

$$\begin{aligned}L_t^u V(t,x)+\frac{1}{2}\|u\|^2&=\nabla_x V(t,x)^\top b(t,x)+\frac{1}{2}\text{Tr}\big(\sigma(t,x)\sigma(t,x)^\top\nabla_x^2 V(t,x)\big)\\&\quad+\frac{1}{2}\left\|u+\sigma(t,x)^\top\nabla_x V(t,x)\right\|^2-\frac{1}{2}\left\|\sigma(t,x)^\top\nabla_x V(t,x)\right\|^2.\end{aligned} \tag{55}$$

The minimum in (53) is therefore attained at

$$u^*(t,x)=-\sigma(t,x)^\top\nabla_x V(t,x),$$

which yields the stated expression for the infimum and completes the proof. $\square$

## F. Proof of Proposition 5.1

We restate the result.

**Proposition F.1.** *Consider the stochastic optimal control problem with cost functional (19) and controlled SDE (20). Let $V$ denote the value function defined in (21), and consider the exponential transformation $\tilde{h}(t,x)=\exp(-V(t,x))$. Then, the function $\tilde{h}$ solves the backward Kolmogorov equation (8) with terminal condition $\tilde{h}(T,x)=\tilde{h}_T(x)$. Moreover, the optimal control is given by*

$$u^*(t,x)=\bar{g}(t)\nabla_x\log\tilde{h}(t,x), \tag{22}$$

*and the resulting closed-loop dynamics satisfy*

$$dY_t=\big(\bar{f}_\theta(t,Y_t)+\bar{g}(t)u^*(t,Y_t)\big)\,dt+\bar{g}(t)\,d\bar{W}_t. \tag{23}$$

*This closed-loop SDE coincides with the guided reverse SDE (13) constructed via approximate Doob's h-transform using the function $\tilde{h}$.*

*Proof.* Given the controlled SDE (20) and the cost functional (47), Corollary E.3 shows that the value function $V(t,x)$ solves the HJB equation

$$\frac{\partial V}{\partial t}(t,x)+\bar{L}_t V(t,x)-\frac{1}{2}\bar{g}(t)^2\|\nabla_x V(t,x)\|^2=0, \tag{56}$$

with the terminal condition

$$V(T,x)=-\log\tilde{h}_T(x). \tag{57}$$

Then, we define $\tilde{h}(t,x)$ by the exponential transformation

$$\tilde{h}(t,x)=\exp(-V(t,x)). \tag{58}$$

Then, we have

$$\partial_t V(t,x)=-\frac{\partial_t\tilde{h}(t,x)}{\tilde{h}(t,x)},\nabla_x V(t,x)=-\frac{\nabla_x\tilde{h}(t,x)}{\tilde{h}(t,x)},\nabla_x^2 V(t,x)=-\frac{\nabla_x^2\tilde{h}(t,x)}{\tilde{h}(t,x)}+\frac{\nabla_x\tilde{h}(t,x)\nabla_x\tilde{h}(t,x)^T}{\tilde{h}(t,x)^2}. \tag{59}$$

Inserting these terms into equation (56), we have

$$\begin{aligned}&-\frac{\partial_t\tilde{h}(t,x)}{\tilde{h}(t,x)}-\frac{\nabla_x\tilde{h}(t,x)}{\tilde{h}(t,x)}\bar{f}_\theta+\frac{1}{2}\bar{g}(t)^2\left\{-\frac{\nabla_x^2\tilde{h}(t,x)}{\tilde{h}(t,x)}+\frac{\nabla_x\tilde{h}(t,x)\nabla_x\tilde{h}(t,x)^T}{\tilde{h}(t,x)^2}\right\}-\frac{1}{2}\bar{g}(t)^2\frac{\nabla_x\tilde{h}(t,x)\nabla_x\tilde{h}(t,x)^T}{\tilde{h}(t,x)^2}\\&=-\frac{\partial_t\tilde{h}(t,x)}{\tilde{h}(t,x)}-\frac{\nabla_x\tilde{h}(t,x)}{\tilde{h}(t,x)}\bar{f}_\theta-\frac{1}{2}\bar{g}(t)^2\frac{\nabla_x^2\tilde{h}(t,x)}{\tilde{h}(t,x)}=0\end{aligned} \tag{60}$$

When $\tilde{h}$ is given by (58), $\tilde{h}(t,x)$ is positive on $[0,T] \times \mathbb{R}^n$. Therefore, multiplying both sides by $\tilde{h}(t,x)$ in equation (60), we have

$$-\partial_t \tilde{h}(t,x) - \nabla_x \tilde{h}(t,x) \bar{f}_\theta - \frac{1}{2} \bar{g}(t)^2 \nabla_x^2 \tilde{h}(t,x) = 0. \tag{61}$$

This implies that the function $\tilde{h}$ solves the linear PDE

$$\frac{\partial \tilde{h}}{\partial t}(t,x) + \bar{L}_t \tilde{h}(t,x) = 0, \tag{62}$$

with the terminal condition

$$\tilde{h}(T,x) = \tilde{h}_T(x). \tag{63}$$

This completes the proof. □

## G. Proof of Theorem 5.2

In this section, we provide the proof of Theorem 5.2.

The argument relies on a known result from path integral control theory (Thijssen & Kappen, 2015), which we briefly recall here for completeness.

Consider the stochastic optimal control problem defined by the cost functional (47) and the controlled SDE (46). Rather than solving the nonlinear HJB equations (49) and (51), path integral control provides an alternative characterization of the optimal control in terms of weighted expectations over trajectories.

The following theorem is adapted from Thijssen & Kappen (2015).

**Theorem G.1** (Path integral control (Thijssen & Kappen, 2015)). *Consider the stochastic optimal control problem with cost functional (47) and controlled SDE (46). Then, the optimal control $u^*$ minimizing (47) admits the representation*

$$u^*(t,x) = u(t,x) + \lim_{\Delta t \to 0} \frac{1}{\Delta t} \frac{\mathbb{E}\left[\exp(-S^u(t)) \int_t^{t+\Delta t} dW_s \,\middle|\, Z_t = x\right]}{\mathbb{E}[\exp(-S^u(t)) \,|\, Z_t = x]}, \tag{64}$$

*where $u$ is an arbitrary baseline control, and the functional $S^u(t)$ is defined by*

$$S^u(t) = \int_t^T \frac{1}{2} \|u(s, Z_s)\|^2 ds + \int_t^T u(s, Z_s)^\top dW_s + \varphi(Z_T). \tag{65}$$

*Here, the conditional expectations are taken with respect to trajectories of (46) generated under the baseline control $u$.*

The proof of Theorem 5.2 follows directly from Theorem G.1 by substituting the controlled SDE (46) with (20), and replacing the terminal cost $\varphi(x)$ with $-\log \tilde{h}_T(x)$.

## H. Proof of Theorem 4.3

We show the proof of Theorem 4.3 giving the upper bound of the relative entropy between the marginal distributions of the ideal guided reverse diffusion process and the approximate guided reverse diffusion process.

We first restate the theorem.

**Theorem 4.3.** *Consider the ideal guided reverse diffusion process defined by the SDE (12), where the guidance is constructed via Doob's h-transform using a function $h$ satisfying the backward Kolmogorov equation (8) with terminal condition $h(T,x) = \mathbf{1}_S(x)$. Consider also the approximate guided reverse diffusion process defined by the SDE (13), where the guidance is constructed using an approximate function $\tilde{h}$. Suppose that Assumption 4.2 holds. Then the relative entropy between the marginal distributions at time $T$ satisfies*

$$D_{\mathrm{KL}}(\mu_T^h \| \mu_T^{\tilde{h}}) \le \frac{1}{2} \rho \tag{17}$$

*where $\rho$ is defined in (16).*

*Proof.* We first recall that the ideal guided reverse diffusion process is given by

$$dY_t^h = \left(\bar{f}_\theta(t, Y_t^h) + \bar{g}(t)^2 \nabla \log h(t, Y_t^h)\right) dt + \bar{g}(t) d\bar{B}_t, \tag{66}$$

where the function $h$ solves the backward Kolmogorov equation with terminal condition $h(T, x) = \mathbf{1}_S(x)$, and $\{\bar{B}_t\}_{t \in [0,T]}$ is a Brownian motion. We next consider the approximate guided reverse diffusion process

$$dY_t^{\tilde{h}} = \left(\bar{f}_\theta(t, Y_t^{\tilde{h}}) + \bar{g}(t)^2 \nabla \log \tilde{h}(t, Y_t^{\tilde{h}})\right) dt + \bar{g}(t) d\bar{W}_t, \tag{67}$$

where $\tilde{h}$ denotes an approximation of $h$, and $\{\bar{W}_t\}_{t \in [0,T]}$ is a Brownian motion.

Let $\mathbb{P}^h$ and $\mathbb{P}^{\tilde{h}}$ denote the path measures induced by the processes $\{Y_t^h\}_{t \in [0,T]}$ and $\{Y_t^{\tilde{h}}\}_{t \in [0,T]}$, respectively. By Girsanov's theorem, under the integrability condition in Assumption 4.2, the Radon–Nikodym derivative between the path measures $\mathbb{P}^h$ and $\mathbb{P}^{\tilde{h}}$ is given by

$$\frac{d\mathbb{P}^h}{d\mathbb{P}^{\tilde{h}}} = \exp\left(-\int_0^T \bar{g}(t) \left(\nabla \log \tilde{h}(t, Y_t^h) - \nabla \log h(t, Y_t^h)\right)^\top d\bar{B}_t + \frac{1}{2} \int_0^T \left\| \bar{g}(t) \left(\nabla \log \tilde{h}(t, Y_t^h) - \nabla \log h(t, Y_t^h)\right) \right\|^2 dt\right), \tag{68}$$

where $\{\bar{B}_t\}_{t \in [0,T]}$ is the Brownian motion appearing in the SDE for $Y^h$ under $\mathbb{P}^h$.

Since the first term in the exponent is a stochastic integral with zero expectation under $\mathbb{P}^h$, the relative entropy between the path measures satisfies

$$\begin{aligned}
D_{\mathrm{KL}}(\mathbb{P}^h \,\|\, \mathbb{P}^{\tilde{h}}) &= \mathbb{E}_{\mathbb{P}^h}\left[\log \frac{d\mathbb{P}^h}{d\mathbb{P}^{\tilde{h}}}\right] \\
&= \frac{1}{2} \int_0^T \mathbb{E}_{\mathbb{P}^h}\left[\left\| \bar{g}(t) \left(\nabla \log \tilde{h}(t, Y_t^h) - \nabla \log h(t, Y_t^h)\right) \right\|^2\right] dt.
\end{aligned} \tag{69}$$

Finally, by the data processing inequality and recalling the notation $\mu_T^h = \mathcal{L}(Y_T^h)$ and $\mu_T^{\tilde{h}} = \mathcal{L}(Y_T^{\tilde{h}})$, we obtain

$$\begin{aligned}
D_{\mathrm{KL}}(\mu_T^h \,\|\, \mu_T^{\tilde{h}}) &\leq D_{\mathrm{KL}}(\mathbb{P}^h \,\|\, \mathbb{P}^{\tilde{h}}) \\
&= \frac{1}{2} \int_0^T \mathbb{E}_{\mathbb{P}^h}\left[\left\| \bar{g}(t) \left(\nabla \log \tilde{h}(t, Y_t^h) - \nabla \log h(t, Y_t^h)\right) \right\|^2\right] dt.
\end{aligned} \tag{70}$$

This completes the proof. $\square$

**Theorem 4.4.** *Under the same assumptions as in Theorem 4.3, if $\rho < 2$, then the success probability under the approximate guided process satisfies*

$$\mu_T^{\tilde{h}}(S) \geq 1 - \sqrt{\frac{1}{2}\rho}. \tag{18}$$

*Proof.* Recall the notations $\mu_T^h = \mathcal{L}(Y_T^h)$ and $\mu_T^{\tilde{h}} = \mathcal{L}(Y_T^{\tilde{h}})$ denoting the marginal distributions at time $T$ of the solutions to the SDEs (66) and (67), respectively.

By Pinsker's inequality, the total variation distance between these measures is bounded in terms of the relative entropy as

$$\left\| \mu_T^h - \mu_T^{\tilde{h}} \right\|_{\mathrm{TV}} \leq \sqrt{\frac{1}{2} D_{\mathrm{KL}}\left(\mu_T^h \,\|\, \mu_T^{\tilde{h}}\right)}. \tag{71}$$

Here, $\|\cdot\|_{\mathrm{TV}}$ denotes the total variation distance.

By definition of the total variation distance, for any measurable set $A$ we have

$$\left| \mu_T^h(A) - \mu_T^{\tilde{h}}(A) \right| \leq \left\| \mu_T^h - \mu_T^{\tilde{h}} \right\|_{\mathrm{TV}}. \tag{72}$$

Combining (71) and (72), and substituting the relative entropy bound provided by Theorem 4.3, we obtain

$$\left|\mu_T^h(A) - \mu_T^{\tilde{h}}(A)\right| \leq \sqrt{\frac{1}{2}D_{\mathrm{KL}}\left(\mu_T^h \,\|\, \mu_T^{\tilde{h}}\right)} \leq \sqrt{\frac{1}{2}\rho}, \tag{73}$$

where $\rho$ is given by (16). Consequently,

$$\mu_T^{\tilde{h}}(A) \geq \mu_T^h(A) - \sqrt{\frac{1}{2}\rho}. \tag{74}$$

By construction of the true guided reverse diffusion process, which is conditioned on reaching the target set $S$ at time $T$, we have $\mu_T^h(S) = 1$. Therefore,

$$\mu_T^{\tilde{h}}(S) \geq 1 - \sqrt{\frac{1}{2}\rho}. \tag{75}$$

which completes the proof. $\qquad\square$

## I. Discussion on Scope of Analysis for Guided Diffusion Models via Doob's $h$-Transform

In this section, we discuss the scope of our theoretical analysis and possible extensions to more general settings, including score approximation error and time-discretization error.

As discussed at the end of Section 4.1, the error analysis in Theorems 4.3 and 4.4 focuses on the error between the ideal guided reverse diffusion process and the approximate guided reverse diffusion process, where the guidance terms are designed by Doob's $h$-transform using the function $h$ and its approximation $\tilde{h}$, respectively. The aim of the analysis is to provide a theoretical characterization of the error induced by approximating the function $h$ in Doob's $h$-transform when implementing the guidance term.

Recent studies (Lee et al., 2022; Chen et al., 2023; Berner et al., 2026) on diffusion models consider score approximation error and time-discretization error, which are not covered in our analysis. The following provides an outline of how our analysis may be combined with these results to account for these components. For example, Chen et al. (2023) derive error bounds for score approximation error and time-discretization error in the reverse diffusion process without guidance. Their analysis uses the relative entropy between the distributions of the ideal reverse diffusion process and the approximate reverse diffusion process induced by score approximation and time discretization, and provides error bounds in terms of the score approximation error and the time-discretization step size.

In our analysis, the following quantity plays a key role in the relative-entropy bound between the ideal guided reverse diffusion process and the approximate guided reverse diffusion process:

$$\rho = \int_0^T \mathbb{E}_{\mathbb{P}^h}\left[\left|\bar{g}(t)\left(\nabla\log\tilde{h}(t, Y_t^h) - \nabla\log h(t, Y_t^h)\right)\right|^2\right] dt. \tag{76}$$

This quantity can be interpreted as measuring the discrepancy between the ideal guidance term designed by $h$ and the approximate guidance term designed by $\tilde{h}$, evaluated along the ideal guided process. The term $\nabla\log\tilde{h}(t, Y_t^h) - \nabla\log h(t, Y_t^h)$ represents the difference between these two guidance functions at time $t$. In principle, the analyses of score approximation error and time-discretization error in Chen et al. (2023) can be combined with our analysis by treating these errors as additional sources of discrepancy between the implemented dynamics and the ideal guided dynamics. From this viewpoint, the total error may be conceptually viewed as involving the guidance-design error together with additional terms corresponding to score approximation and time discretization, for example,

$$\int_0^T \mathbb{E}_{\mathbb{P}^h}\left[\left|\bar{g}(t)\left(\nabla\log\tilde{h}(t, Y_t^h) - \nabla\log h(t, Y_t^h)\right) + \Delta_{\mathrm{score}} + \Delta_{\mathrm{disc}}\right|^2\right] dt, \tag{77}$$

where $\Delta_{\mathrm{score}}$ and $\Delta_{\mathrm{disc}}$ denote error terms corresponding to score approximation and time discretization, respectively.

Under appropriate assumptions, such as those considered in Chen et al. (2023), bounds on score approximation error and time-discretization error may be incorporated into our KL-based analysis as additional discrepancy terms. Since the guarantee in Theorem 4.4 is controlled by the value of $\rho$, this suggests a possible route to obtaining probabilistic guarantees for the implemented guided diffusion process by replacing the guidance-only discrepancy with an overall discrepancy that also accounts for score approximation and time discretization.

When the guidance function is computed by path integral control, Monte Carlo sampling is used to estimate the guidance function, which introduces sampling error. To evaluate this error, one can use sample complexity analyses for path integral control. Such analyses have been studied in the literature on sample complexity for path integral control; see, for example, Patil et al. (2024).

## J. Algorithm Details of Guided Diffusion Models via Path Integral Control

Consider the reverse-time SDE with the guidance,

$$dY_t^u = \left(-f(T-t, Y_t^u) + \sigma(T-t)^2 s_\theta(T-t, Y_t^u) + \sigma(T-t)u(t, Y_t^u)\right)dt + \sigma(T-t)d\bar{W}_t, \tag{78}$$

where $u : [0, T] \times \mathbb{R}^n \to \mathbb{R}^n$ is the guidance term.

We assume that the score network $s_\theta$ is trained by minimizing the score matching loss function, which we show here again,

$$L(\theta) = \mathbb{E}_{t\sim\mathcal{U}(0,T)}\left[g(t)^2\mathbb{E}_{X_0,X_t}\left[\|s_\theta(t, X_t) - \nabla_x\log p(X_t \mid X_0)\|^2\right]\right], \tag{79}$$

where $\mathcal{U}(0, T)$ denotes the uniform distribution on $(0, T)$, $p(X_t \mid X_0)$ is the transition density of the forward SDE (1) given $X_0$, and the expectation is taken over $X_0 \sim \mathcal{L}_{\text{data}}$ and the corresponding solution $X_t$ of the SDE (1)

After the training, the guidance function is obtained by the solving the stochastic optimal control problem given by

$$\min_u \mathbb{E}\left[\int_0^T \frac{1}{2}\|u(t, Y_t^u)\|^2 dt - \log\tilde{h}_T(Y_T^u)\right], \tag{80}$$

subject to the controlled SDE given by (78), where $\tilde{h}_T : \mathbb{R}^n \to \mathbb{R}_{>0}$ is a positive function representing the terminal condition.

The optimal control problem defined by (80) and (78) is solved by the path integral control. As shown in Theorem 5.2, the optimal controller is given by

$$u^*(t, x) = \bar{u}(t, x) + \lim_{\Delta t \to 0} \frac{1}{\Delta t} \frac{\mathbb{E}\left[\exp\left(-S^{\bar{u}}(t)\right)\int_t^{t+\Delta t} d\bar{W}_s \mid Y_t = x\right]}{\mathbb{E}\left[\exp\left(-S^{\bar{u}}(t)\right) \mid Y_t = x\right]}, \tag{81}$$

where $u$ is an arbitrary control function, and $S^u(t)$ is the cost functional defined by

$$S^{\bar{u}}(t) = \int_t^T \frac{1}{2}\|\bar{u}(s, Y_s)\|^2 ds + \int_t^T \bar{u}(s, Y_s)^T d\bar{B}_s - \log\tilde{h}_T(Y_T), \tag{82}$$

where $\bar{u}$ is an arbitrary baseline control usually implemented by iLQG, the conditional expectations are taken with respect to trajectories generated by (20) under $u$, $S^u(t)$ is defined in (82), where $-\log\tilde{h}_T(Y_T)$ corresponds to the terminal cost function of the optimal control problem.

In the implementation, the controller is implemented via the discrete-time approximation: at the time $t_i$,

$$u^*(t_i, x)\Delta t = \bar{u}(t_i, x)\Delta t + \frac{\mathbb{E}\left[\exp\left(-S^{\bar{u}}(t_i)\right)\Delta\bar{W}_{t_i} \mid Y_{t_i} = x\right]}{\mathbb{E}\left[\exp\left(-S^{\bar{u}}(t_i)\right) \mid Y_{t_i} = x\right]}, \tag{83}$$

where $\Delta t$ is the time step size, and $\Delta\bar{W}_{t_i}$ is the increment of the Brownian motion $\bar{W}_s$, that is, $\Delta W_{t_i} = \bar{W}_{t_{i+1}} - \bar{W}_{t_i}$. The integration $S^u$ in (82) is also computed by discrete-time approximation

$$S^{\bar{u}}(t_i) = \sum_{j=i}^{N_T-1}\left(\frac{1}{2}\|\bar{u}(t_j, Y_{t_j})\|^2\Delta t + \bar{u}(t_j, Y_{t_j})^T\Delta\bar{W}_{t_j}\right) - \log\tilde{h}_T(Y_{N_T}), \tag{84}$$

where $N_T = \lfloor T/\Delta t\rfloor$. The discretized path of $Y_{t_i}$ is approximated by the Euler-Maruyama method applied to (20):

$$Y_{t_{i+1}}^{\bar{u}} = Y_{t_i}^{\bar{u}} + \left(-f(T-t_i, Y_{t_i}^{\bar{u}}) + \sigma(T-t_i)^2 s_\theta(T-t_i, Y_{t_i}^{\bar{u}}) + \sigma(T-t_i)\bar{u}(t_i, Y_{t_i}^{\bar{u}})\right)\Delta t + \sigma(T-t_i)\Delta\bar{W}_{t_i}, \tag{85}$$

where $\bar{u}$ is the baseline control typically implemented by iLQG. The conditional expectations in (83) are approximated by the Monte Carlo method, that is,

$$\mathbb{E}\left[\exp\left(-S^{\bar{u}}(t_i)\right)\Delta\bar{W}_{t_i} \mid Y_{t_i} = x\right] \approx \frac{1}{L}\sum_{l=1}^{L}\exp\left(-S^{\bar{u},l}(t_i)\right)\Delta\bar{W}_{t_i}^l, \tag{86}$$

and

$$\mathbb{E}\left[\exp\left(-S^{\bar{u}}(t_i)\right) \mid Y_{t_i} = x\right] \approx \frac{1}{L}\sum_{l=1}^{L}\exp\left(-S^{\bar{u},l}(t_i)\right), \tag{87}$$

where $\{S^{\bar{u},l}(t_i)\}_{l=1,\dots,L}$ are computed by simulating $L$ trajectories $\{Y_{t_j}^{\bar{u},l}\}_{j=i,\dots,N_T}^{l=1,\dots,L}$ of the controlled SDE (85) with $Y_{t_i}^{\bar{u},l} = x$ for each $l$ using the baseline control $\bar{u}$, and evaluating $S^{\bar{u},l}(t_i)$ by (84).

---

**Algorithm 2** Algorithm of Path Integral Control for Guidance

---

1: Set time step size $\Delta t$, total number of time steps $N_T = \lfloor T/\Delta t \rfloor$, and numbers of particles $K$ and $L$.
2: Sample initial particles $\left\{\tilde{Y}_0^{\bar{u},k}\right\}_{k=1,\dots,K}$ from the initial distribution $P_{\text{noise}}$.
3: **for** $i = 0, 1, 2, \dots, N_T$ **do**
4:   Set $t = t_i := i\Delta t$.
5:   Find a baseline control $\bar{u}$ that approximately minimize (80) subject to (78) using an algorithm such as iLQG and compute discretized paths $\{\tilde{Y}_{t_j}^{\bar{u},k}\}_{j=0,\dots,i}^{k=1,\dots,K}$ by the discretized equation of (85) with the suboptimal control $\bar{u}$.
6:   Compute controls $u^*(t_i, Y_{t_i}^{\bar{u},k})$ for particles $\{Y_{t_i}^{\bar{u},k}\}_{k=1,\dots,K}$ using (83)
7:   Apply control $u^*(t_i, Y_{t_i}^{\bar{u},k})$ to the control system (78).
8: **end for**

---

## K. Details of Robotic System Planning

We consider the planning problem for robotic systems within the framework of diffusion-based models.

We assume that the system dynamics are given by a discrete-time controlled dynamical system. Let $s_k \in \mathbb{R}^{n_s}$ denote the state and $a_k \in \mathbb{R}^{n_a}$ the control input at time step $k \in \{0, 1, \dots, T-1\}$. The dynamics are described by

$$s_{k+1} = F(s_k, a_k), \tag{88}$$

where $F : \mathbb{R}^{n_s} \times \mathbb{R}^{n_a} \to \mathbb{R}^{n_s}$ denotes the (possibly unknown) state transition function.

This study follows the diffusion-based planning via imitation learning framework of Janner et al. (2022) without considering reward. Instead of training the reward network, our study attempts to find a stragedy to build a guidance function for robot planning when the condition is specified.

The diffusion-based planning framework of Janner et al. (2022) models the distribution of expert trajectories using diffusion models. We assume that the diffusion model is trained on a dataset of expert demonstrations and learns the distribution over trajectories $\tau$.

In the forward process (1) and the approximate reverse process (5) with the score network, each trajectory

$$\tau = (s_0, a_0, s_1, a_1, \dots, s_{T-1}, a_{T-1}, s_T)$$

is treated as a data point in $\mathbb{R}^{(n_s+n_a)T+n_s}$ by concatenating all state and action variables.

Accordingly, sampling a trajectory corresponds to simulating the reverse SDE (5) from the initial distribution $\mathcal{L}_{\text{noise}}$, and the generated trajectory is given by $\tau = Y_T$.

Under this identification, the forward SDE can be written as

$$d\tau_t = f(t, \tau_t)\,dt + g(t)\,dB_t, \tag{89}$$

where $\tau_t \in \mathbb{R}^{(n_s+n_a)T+n_s}$ denotes the trajectory-valued state at time $t$, and the drift $f$ and diffusion coefficient $g$ are defined as in Section 3.1. The corresponding reverse SDE is given by

$$d\tau_t = \left(-f(T-t, \tau_t) + g(T-t)^2 s_\theta(T-t, \tau_t)\right) dt + g(T-t)d\bar{B}_t, \tag{90}$$

where $s_\theta : [0, T] \times \mathbb{R}^{(n_s+n_a)T+n_s} \to \mathbb{R}^{(n_s+n_a)T+n_s}$ denotes the score network.

Given a dataset of expert trajectories, the score network is trained by minimizing the score matching loss (79).

We now apply the proposed guidance framework to the diffusion-based planning setting.

Given a prescribed target (or safe) set $S \subset \mathbb{R}^{(n_s+n_a)T+n_s}$, the Doob $h$-transform suggests using the terminal function

$$h_T(\tau) = \mathbf{1}_S(\tau), \tag{91}$$

where $\mathbf{1}_S$ denotes the indicator function of the set $S$. As discussed earlier, the discontinuity of this terminal condition motivates the use of a smooth approximation. Specifically, we introduce a strictly positive terminal function $\tilde{h}_T : \mathbb{R}^{(n_s+n_a)T+n_s} \to \mathbb{R}_{>0}$ defined by

$$\tilde{h}_T(\tau) = \exp\left(-\frac{1}{\lambda}\text{dist}(\tau, S)^2\right), \tag{92}$$

where the distance to the set $S$ is given by

$$\text{dist}(\tau, S) = \inf_{y \in S} \|\tau - y\|, \tag{93}$$

and $\lambda > 0$ is a tuning parameter controlling the sharpness of the approximation.

Given the terminal function $\tilde{h}_T$ and the trained score network $s_\theta$, the guidance is obtained by solving the stochastic optimal control problem

$$J(u) = \mathbb{E}\left[\int_0^T \frac{1}{2}\|u(t, \tau_t)\|^2 dt - \log \tilde{h}_T(\tau_T)\right], \tag{94}$$

subject to the controlled dynamics induced by the reverse SDE,

$$d\tau_t = \left(-f(T-t, \tau_t) + g(T-t)^2 s_\theta(T-t, \tau_t) + g(T-t)u(t, \tau_t)\right) dt + g(T-t)d\bar{B}_t. \tag{95}$$

Let $V(t, \tau)$ denote the value function associated with this optimal control problem. By Proposition 5.1, the optimal guidance function is then given by

$$u^*(t, \tau) = -g(T-t)\nabla_\tau V(t, \tau). \tag{96}$$

An algorithmic implementation of the planning method based on the stochastic optimal control problem (94)–(95) is provided in Appendix 3.

---

**Algorithm 3** Guided Planning via Stochastic Optimal Control

---

1: Set planning horizon $T$ and target set $S$.
2: Measure the initial system state $s_0$ of (88) and set $\hat{s}_0 = s_0$.
3: Generate a trajectory $\hat{\tau} = (\hat{s}_0, \hat{a}_0, \hat{s}_1, \hat{a}_1, \dots, \hat{s}_{T-1}, \hat{a}_{T-1}, \hat{s}_T)$ by simulating the guided reverse SDE (95) with guidance (96) from the initial distribution $P_{\text{noise}}$.
4: **for** $k = 0, 1, \dots, T-1$ **do**
5:     Apply the planned action $a_k = \hat{a}_k$ to the system (88).
6: **end for**

---

Furthermore, as shown in Theorem 5.2, the optimal guidance function can alternatively be computed via the path integral control formulation.

Introducing the functional $S^{\bar{u}}(t)$ defined by

$$S^{\bar{u}}(t) = \int_t^T \frac{1}{2}\|\bar{u}(s, \tau_s)\|^2 ds + \int_t^T \bar{u}(s, \tau_s)^\top d\bar{B}_s - \log \tilde{h}_T(\tau_T), \tag{97}$$

where $\bar{u}$ denotes an arbitrary baseline control, the optimal guidance function is given by

$$u^*(t, \tau) = \bar{u}(t, \tau) + \lim_{\Delta t \to 0} \frac{1}{\Delta t} \frac{\mathbb{E}\left[\exp\left(-S^{\bar{u}}(t)\right) \int_t^{t+\Delta t} d\bar{B}_s \mid \tau_t = \tau\right]}{\mathbb{E}\left[\exp\left(-S^{\bar{u}}(t)\right) \mid \tau_t = \tau\right]}. \tag{98}$$

Here, $\tau_t$ denotes the solution of the controlled SDE (95) at time $t$.

An algorithmic implementation of the path integral control–based planning method is summarized in Algorithm 4. Implementation details are provided in Section J.

---

**Algorithm 4** Guided Planning via Path Integral Control (Open-loop)

---
1: Set planning horizon $T$ and target set $S$.
2: Measure the initial system state $s_0$ of (88) and set $\hat{s}_0 = s_0$.
3: Generate a trajectory $\hat{\tau} = (\hat{s}_0, \hat{a}_0, \hat{s}_1, \hat{a}_1, \ldots, \hat{s}_{T-1}, \hat{a}_{T-1}, \hat{s}_T)$ by simulating the guided reverse SDE (95) from the initial distribution $P_{\text{noise}}$, where the guidance is computed via the path integral control formula (98).
4: **for** $k = 0, 1, \ldots, T-1$ **do**
5:    Apply the planned action $a_k = \hat{a}_k$ to the system (88).
6: **end for**

---

## L. Details of Numerical Experiments

### L.1. Experimental Parameters

Most experiments are conducted in a 2D navigation domain with state space $\mathbb{R}^2$. The diffusion model is trained via imitation learning on expert trajectories generated by A* planning with trajectory smoothing (Hart et al., 1968). Key hyperparameters are summarized in Table 3. We further evaluate generalization beyond point-mass dynamics to higher-dimensional robotic systems with a 7-DOF Franka Panda robot.

| Parameter | Value |
|---|---|
| PIC rollouts ($L$) | 64 |
| Rollout horizon | Full ($T - t_{\text{current}}$) |
| Diffusion timesteps ($T$) | 150 |
| Goal Weight ($w_{\text{goal}}$) | 50.0 |
| Constraint Weight ($w_{\text{obs}}$) | 100.0 |
| PIC Temperature ($\alpha$) | 1.0 |
| Score network | ScoreNetworkV2 (hidden_dim$= 512$, 8 blocks, 8.7M parameters) |
| Number of trials | 100 (5 seeds $\times$ 20 trials) |

*Table 3.* Hyperparameters for numerical experiments.

**2D Navigation (Point-Mass).** We evaluate on three benchmark scenarios of increasing difficulty (see Figure 1):

- **Simple**: Single obstacle at origin with radius 0.8. Start: $(-3, 0)$, Goal: $(3, 0)$.

- **Medium**: Three obstacles: Center (radius 0.8), Right Flank (radius 0.6), Left Flank (radius 0.5). Start: $(-2, 2)$, Goal: $(2, -2)$.

- **HardTrap**: Four obstacles forming a trap configuration requiring precise navigation through narrow passages. Start: $(-3, 0)$, Goal: $(3, 0)$.

**7-DOF Franka Panda (Zero-Shot Transfer).** Three scenarios in a 3D end-effector workspace evaluated via zero-shot transfer from the 2D-trained model. Start: $(0.3, 0.0, 0.5)$, Goal: $(0.7, 0.0, 0.5)$ across all scenarios.

- **Wall**: A planar barrier at $x = 0.5$, $y \in [-0.3, 0.3]$, $z \in [0.2, 0.8]$, represented as 4 spheres of radius 0.08.

| Target $n$ | PIC (Ours) / Efficiency Ratio | RS |
|:---:|:---:|:---:|
| 10 | 12.0/1.20× | >5000 (timeout) |
| 50 | 53.0/1.06× | >5000 (timeout) |
| 100 | 112.0/1.12× | >5000 (timeout) |

*Table 4.* Sample complexity comparison between PIC and RS. We report the average number of sampling attempts required to obtain $n$ valid constraint-satisfying trajectories. RS fails to produce valid samples within the allotted budget.

- **Corridor**: Two parallel walls creating a narrow passage, represented as 6 spheres of radius 0.06 along $y = \pm 0.2$.

- **Dense Forest**: Six small spheres of radius 0.06–0.08 placed in the end-effector workspace at (0.6–1.0, 0.4–1.0, 0.3–0.6). This is distinct from the 2D mega-benchmark Dense Forest scenario.

**Large-Scale Mega-Benchmark (2D Point-Mass).** Five topological scenarios designed to test worst-case constraint satisfaction in highly non-convex environments:

- **Dense Forest**: Five circular obstacles in $[-4, 4]^2$ requiring navigation through tight gaps.

- **Narrow Corridor**: A single narrow passage between obstacle walls.

- **U-Trap**: A U-shaped obstacle configuration requiring the agent to avoid local minima.

- **Barrier Wall**: A wall obstacle blocking the direct path to the goal.

- **Random Scatter**: Randomly placed obstacles across the workspace.

### L.2. Sample Complexity Analysis

This experiment evaluates the ability of the proposed technique to efficiently generate constraint-satisfying trajectories (see Table 4). Here, we compare PIC guidance against rejection sampling (RS), a common baseline that repeatedly samples trajectories from the unguided diffusion model until feasibility is achieved. Due to the vanishing probability of sampling feasible trajectories in constrained environments, RS fails to obtain valid solutions within a practical sampling budget. In contrast, PIC exhibits near-optimal sample efficiency: generating 100 feasible trajectories requires only approximately 112 sampling attempts on average. These results indicate that the proposed path integral formulation enables efficient constraint satisfaction without incurring the prohibitive sampling costs associated with rejection-based approaches.

### L.3. Reflected Diffusion Baseline

We implement the reflected diffusion baseline following Lou & Ermon (2023). The key components are:

**Reflected Brownian Motion.** The forward process uses reflected Brownian motion on a box domain $[\ell, u]^d$:

$$dX_t = \sigma \, dB_t, \quad X_t \in [\ell, u]^d, \tag{99}$$

with instantaneous reflection at the boundaries. Reflection is implemented using the "fold" method: if $x < \ell$, the reflected value is $\ell + (\ell - x)$.

**Constrained Denoising Score Matching (CDSM).** The score network $s_\theta(x, t)$ is trained using the CDSM loss:

$$L(\theta) = \mathbb{E}_{t, x_0, x_t} \left[ \| s_\theta(x_t, t) - \nabla_{x_t} \log p(x_t \mid x_0) \|^2 \right], \tag{100}$$

where the transition kernel $p(x_t \mid x_0)$ accounts for boundary reflections.

**Limitation for Obstacle Avoidance.** A fundamental limitation of reflected diffusion for planning is that constraints are *baked into training*. The method learns the boundary structure from the training data but **cannot generalize to novel obstacles not present during training**. This explains why Reflected Diffusion fails to generalize to novel obstacle configurations, as demonstrated in the benchmark results in Table 1. In contrast, our proposed PIC guidance is *training-free* and handles arbitrary constraints specified at inference time.

## L.4. Score Approximation Error

A key assumption underlying our theoretical guarantees (Theorem 4.3) is that the learned score network $s_\theta$ provides a reasonable approximation of the true score $\nabla \log p(t, x)$. We empirically validate this by measuring score approximation quality across diffusion timesteps $t \in [0, T]$ using three metrics: MSE $\|s_\theta - \nabla \log p_t\|^2$, cosine similarity, and relative error (see Figure 5).

The score approximation improves substantially as $t$ increases: cosine similarity rises from $0.12$ at $t = 0.05$ to above $0.99$ at $t \geq 0.75$, while relative error drops from $0.99$ to below $0.12$. This indicates that the score network provides accurate directional guidance during the later stages of the reverse diffusion process, which are most critical for constraint satisfaction.

Importantly, despite non-trivial approximation error at early timesteps, PIC maintains strong empirical safety performance across the benchmark scenarios. This suggests that the path integral guidance can be empirically robust to imperfect score estimation, although our theoretical guarantees isolate the guidance-approximation error and do not explicitly account for score approximation error.

**Score Model Approximation Quality Across Noise Levels**

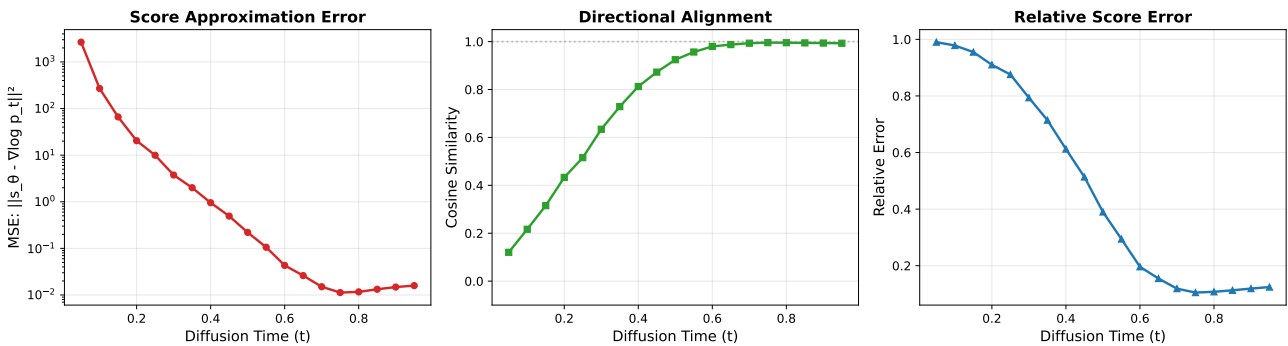

*Figure 5.* Score model approximation quality across diffusion timesteps. Left: MSE decreases as $t$ increases. Center: cosine similarity approaches $1.0$ for $t \geq 0.6$. Right: relative error drops below $0.12$ for $t \geq 0.7$, confirming accurate score estimates during the denoising phase critical for planning.

## L.5. Zero-Shot Transfer: Implementation Details

We validate zero-shot transfer of the 2D-trained model to higher-dimensional robotic systems without any task-specific retraining. We consider a 7-DOF Franka Panda robot as a quantitative benchmark. The PyTorch implementation is provided at https://github.com/shxshank/doob-guided-diffusion

**Forward Kinematics Mapping.** Let $q \in \mathbb{R}^n$ denote the joint configuration of an $n$-DOF arm. The forward kinematics map $\mathrm{FK} : \mathbb{R}^n \to \mathbb{R}^2$ computes the end-effector position:

$$x_{\mathrm{ee}} = \mathrm{FK}(q) = \sum_{i=1}^n L_i \begin{pmatrix} \cos\left(\sum_{j=1}^i q_j\right) \\ \sin\left(\sum_{j=1}^i q_j\right) \end{pmatrix}, \tag{101}$$

where $L_i$ are the link lengths.

**Transfer Mechanism.** The 2D score model captures general trajectory smoothness priors in end-effector space. During inference:

1. Generate candidate trajectories in joint space: $\tau_q = (q_0, q_1, \ldots, q_T)$.

2. Map to end-effector space: $\tau_{\mathrm{ee}} = (\mathrm{FK}(q_0), \ldots, \mathrm{FK}(q_T))$.

3. Evaluate PIC guidance using the 2D score: $u^* = -\nabla_{\tau_{\mathrm{ee}}} V(\tau_{\mathrm{ee}})$.

4. Project guidance back to joint space via the Jacobian: $u_q = J^\dagger(q) \cdot u^*$.

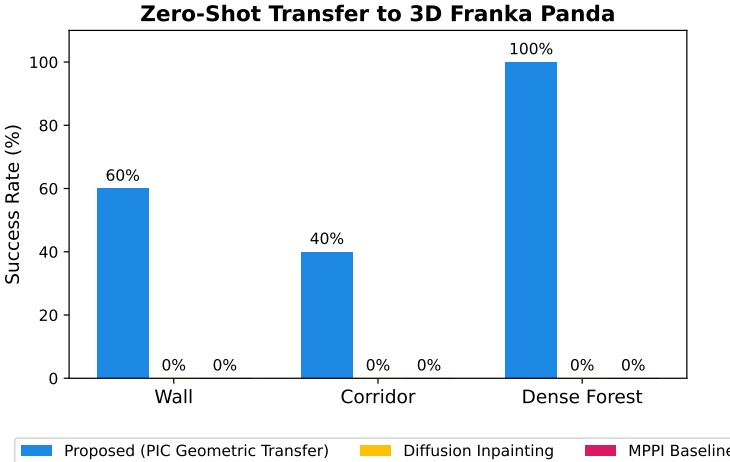

*Figure 6.* Zero-shot transfer to 7-DOF Franka Panda robot across Wall, Corridor, and Dense Forest scenarios. PIC achieves non-zero success in all scenarios while maintaining 100% safety, whereas baseline methods reach the goal in none.

| Method | Dense Forest Safe/Succ | Narrow Corr. Safe/Succ | U-Trap Safe/Succ | Barrier Wall Safe/Succ | Scatter Safe/Succ |
|---|---|---|---|---|---|
| MPPI | 100/0 | 100/0 | 100/0 | 100/0 | 100/0 |
| Gradient | 100/0 | 100/0 | 100/0 | 100/0 | 100/0 |
| Inf. Guidance | 100/0 | 100/40 | 100/0 | 100/40 | 100/40 |
| Diff. Inpainting | 40/40 | 80/80 | 0/0 | 20/20 | 0/0 |
| PIC (Ours) | 100/80 | 100/100 | 100/100 | 100/60 | 100/80 |

*Table 5.* Safety / Success (%) on five topological scenarios. PIC achieves 100% safety across all environments while consistently reaching the goal.

This transfer works because the 2D prior encodes "smoothness" and "goal-reaching" behaviors that are agnostic to the specific kinematic structure. Quantitative results for the 7-DOF Franka Panda are reported in Table 2 and Figure 6.

### L.6. Large-Scale Randomized Benchmark

To further evaluate robustness beyond fixed obstacle configurations, we benchmark PIC against MPPI, Gradient Guidance, Inference Guidance, and Diffusion Inpainting on five challenging topological scenarios designed to test worst-case constraint satisfaction. As shown in Table 5, MPPI and Gradient Guidance achieve high safety but zero success across all scenarios, failing to reach the goal entirely. Diffusion Inpainting succeeds in some scenarios but at the cost of significantly reduced safety. In contrast, PIC achieves 100% safety across all scenarios while maintaining meaningful success rates, demonstrating strong empirical constraint satisfaction in highly non-convex environments. Diffusion Inpainting achieves higher success in the Narrow Corridor scenario, but at the cost of reduced safety (80%) and complete failure in U-Trap and Random Scatter, highlighting its inconsistency across topologies.

### L.7. Language-Guided Planning: Obtaining $\tilde{h}_T$

The language-to-cost compilation pipeline transforms natural language commands into differentiable terminal cost functions $\tilde{h}_T(x)$.

**Template-Based Compilation.** For common constraint types, we use pattern matching to identify the constraint class:

- "Avoid [region]" $\rightarrow \tilde{h}_T(x) = \exp(-\lambda \cdot d_{\min}(x, \mathcal{O})^2)$ (obstacle avoidance)

- "Reach [location]" $\rightarrow \tilde{h}_T(x) = \exp(-\lambda \cdot \|x_T - g\|^2)$ (goal reaching)

- "Stay within [bounds]" $\rightarrow \tilde{h}_T(x) = \exp(-\lambda \cdot \max(0, \|x\| - r)^2)$ (bounded region)

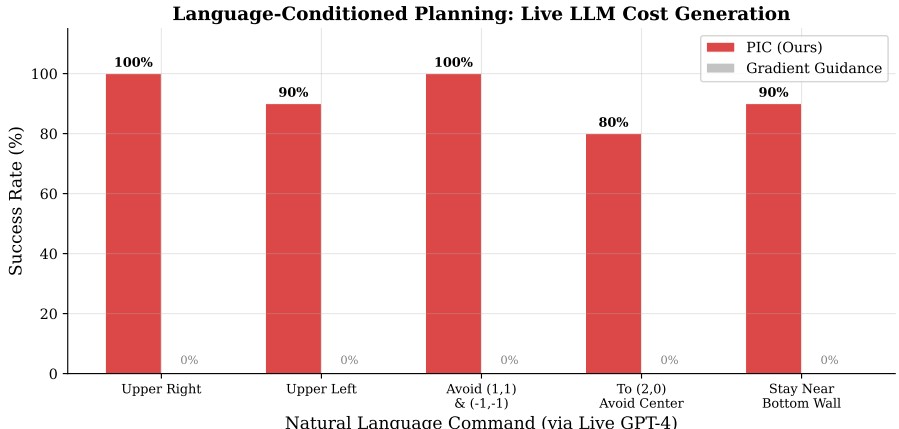

*Figure 7.* Quantitative performance of language-guided zero-shot transfer vs. baseline (Direct PIC).The bar chart illustrates the success rate of the zero-shot transfer across four distinct language-guided tasks in the 3D environment: "Reach Upper Right," "Avoid Red Zone," "Passthrough Narrow," and "Multistep Reach." The Proposed Method (Language-Guided) consistently achieves a 100% success rate across all tasks (green bars), demonstrating the robust generalization of the 2D-trained diffusion prior to 3D control via the geometric mapping layer. In contrast, the Baseline (Direct PIC) fails completely (0% success, red hatched bars) because it lacks the hierarchical planning capability to interpret the high-level language constraints into the correct spatial sub-goals required for the 7-DOF planner.

**LLM-Based Compilation (GPT-4).** For complex or compositional commands, we prompt GPT-4 to generate a Python lambda function representing $\tilde{h}_T$.

**Cost-Function Replacement Baseline.** As an ablation, we evaluate directly using the LLM-generated cost without PIC guidance—simply adding $-\nabla \tilde{h}_T$ to the diffusion drift. This baseline achieves lower success rates because the raw gradient lacks the Monte Carlo importance weighting that PIC provides for navigating complex cost landscapes.

Prompts used for code generation are shown below.

### L.8. Prompts

We use the following system prompt for GPT-4 to compile natural language constraints into differentiable cost functions:

We employ two distinct prompting strategies depending on the planning domain.

**1. Code Generation Query.** For the main pipeline, we interpret constraints into differentiable code. This prompt supports reduced-order models and full kinematic collision checking.

```
You are a robotics code generator. Convert the user's constraint into a PyTorch cost
    function.

Context: {context}
Constraint: "{text}"

The function should return a cost tensor. For robot arm constraints:
- Use link_collision_cost(q, robot, obstacles, safety_margin, weight)
- robot is a RobotArmKinematics object with n_links
- obstacles is a list of {{'center': [x,y], 'radius': r}}

Return ONLY the lambda definition matching:
cost_fn = lambda q: ...

Example for "Keep robot arm 0.1m from obstacles":
cost_fn = lambda q: link_collision_cost(q, robot, obstacles, 0.1, 10.0)
```

**2. Structured Parsing Query (7-DOF Zero-Shot).** For the 7-DOF Franka Panda experiments (Section L.5), we use GPT-4 to parse ambiguous instructions into structured geometric goals and obstacles, which are then passed to the generalized guidance function.

```
You are a robot manipulation assistant. Convert natural language into structured JSON
    for a 7-DOF Franka Panda robot arm task.

The robot operates in a 3D workspace with coordinates in meters:
- X: forward (0 to 0.8)
- Y: left/right (-0.5 to 0.5)
- Z: up (0 to 0.8)

Output format:
{
  "goal": [x, y, z],
  "obstacles": [
    {"center": [x, y, z], "radius": r, "description": "..."},
    ...
  ],
  "interpretation": "Brief explanation of how you interpreted the ambiguous instruction"
}

Guidelines:
- "Reach the goal" means move end-effector to a target position
- "Red zone" / "avoid" means spherical obstacle region
- Use reasonable values: radii 0.05-0.15m, positions within workspace
- Be creative but reasonable in interpreting ambiguous instructions
- Return ONLY valid JSON
```

