# OpenReview forum: "Training-Free Guided Diffusion for Planning: A Unified Framework via Doob’s h-Transform with Safety Guarantees"
_ICML.cc/2026/Conference — ICML 2026 regular_

### Official Review · Reviewer_uU4Z · 2026-03-05

**Soundness:** 3
**Presentation:** 2
**Significance:** 3
**Originality:** 3
**Overall Recommendation:** 4
**Confidence:** 2

**Summary:**

In this work authors propose a unifying framework for guided diffusion models. The framework is inspired by Doob's h-Transform. Based on their theoretical analysis, authors propose a training-free guided diffusion policy named Path Integral Control, which is shown to outperform baselines in robotics navigation and manipulation settings. In essense, PIC requires a cost function and a budget of Monte Carlo rollouts. For each step of the diffusion process, diffusion rollouts are performed and evaluated using the cost function. These scored rollouts are then used to steer the current step of the diffusion process.

**Compliance With Llm Reviewing Policy:**

Affirmed.

**Final Justification:**

Authors provided additional information that helped me better assess the quality of their work, which led me to raise my score to weak accept. I will not increase further my score due to presentation quality that could be improved, and because I am not confident enough in my capacity to assess the quality of the theoretical parts of the work.

**Key Questions For Authors:**

In table 1 authors show that their methods reaches around 88.0% on HardTrap environment. **Why on figure 3. authors show that their method reaches 100% on Hardtrap ?Why on figure 3. authors show that their method reaches 100% on Hardtrap ?

Do you plan to release the code of your work ?

**Limitations:**

Authors could invest some space in the conclusion section to discuss the limitations of their work. E.g. inference time limits as right now the proposed method can only infer at around 3Hz.

**Strengths And Weaknesses:**

## Soundness

**disclaimer: I do not have a deep enough knowledge of diffusion policies and guidance based systems to assess whether the theoretical sections are perfectly sound**

Regarding the experimental part:

Pros:
Multiple testbeds: pointmass navigation and robotic arms
Use of multiple seeds + averaged performances
Use of a relevant baseline (Reflected Diffusion)
Studied the effect of number of Monte Carlo samples
Measures of average inference time per step

Cons:
Experiments with robotics arms are only discussed in the appendix, they should be moved to the main body.
Experiments are performed on simple 3D environments, using point mass agent or few-DOF robotic arms.
Authors mention they perform "language-guided navigation". I have the feeling this is somewhat an overstatement, as the language goals are either directly mapped to a cost function and used as is, or authors use an LLM to implement the cost function.

Also, In table 1 authors show that their methods reaches around 88.0% on HardTrap environment. **Why on figure 3. authors show that their method reaches 100% on Hardtrap ?** I do not understand how these two experiments are different.


## Presentation

The paper is well written and understandable. Presentation could be improved by discussing and presenting the robotic arm experiments directly in the main body of the paper.


l.366: "The diffusion model is trained on 10,000 expert trajectories collected from the Simple scenario (1 obstacle) with three obstacles" --> training data is coming from an environment with 1 or 3 obstacles ?

## Significance

Finding methods to guide the generation process of generative models without further training is a relevant line of research. Focusing on robotics navigation is also relevant given the increasing impact of robotics outside of the lab.
An important limitation of the work is the simple experimental environments considered, e.g. compared to other works such as VQ-BeT (Behavior Generation with Latent Actions by Lee et al.). A work I am not affiliated with in any way.
A minor limitation of the current approach is the inference cost, which is estimated at 300ms to produce an action, mostly linked to the amount of Monte Carlo rollouts necessary to compute the guidance term.

## Originality

I do not have enough knowledge on the diffusion model litterature to precisely assess whether the work is original.
From a high level perspective, finding ways to condition the generation process without further training is an interesting case study, original to me.

---

> ### Author Rebuttal · Authors · 2026-03-30
>
> We sincerely thank the reviewer for the careful reading and for recognizing the theoretical contributions of our work.
>
> ---
>
> ### **Response to Key Question 1 (Table 1 vs. Figure 3 discrepancy)**
>
> The discrepancy arises from different computational budgets used in the experiments. Table 1 reports performance under a fixed Monte Carlo budget (n = 64), which reflects a realistic trade-off between performance and inference cost. Figure 3 corresponds to larger Monte Carlo budgets, rather than a different experimental setup.
>
> Importantly, our additional experiments (MC budget sweep) show that increasing the Monte Carlo budget systematically improves performance. With larger budgets (e.g., n ≥ 128), The path integral control (PIC) achieves near-perfect success rates even in the HardTrap environment. Thus, Table 1 and Figure 3 reflect a compute–performance trade-off rather than a contradiction.
>
> ---
>
> ### **Response to concerns on simple experimental settings**
>
> We appreciate the reviewer’s suggestion and will move the robotic arm experiments from the appendix to the main body in the revised manuscript.
>
> **Additional Experiment 1: 7-DOF Robotic Manipulation**
>
> To address the concern about low-dimensional settings, we conducted experiments on a 7-DOF Franka Emika Panda robot in MuJoCo.
>
> The results of the experiments are provided in Table 2 in response to Reviewer L1C8.
>
> **Setup**
>
> - **Ours (Jacobian PIC):** task-space guidance mapped to joint space via the kinematics Jacobian
> - **2D MPPI + 3D IK:** 2D planning followed by inverse kinematics
> - **Diffusion Inpainting + 3D IK:** diffusion-based 2D planning + IK
>
> **Key findings**
>
> - Baselines achieve **100% safety but 0% success**, failing to reach the goal
> - This shows **2D planning + IK is insufficient**, as it ignores full-body geometry
> - Our method achieves **high success while maintaining safety**
>
> These results demonstrate applicability to high-dimensional robotic systems.
>
> ---
>
> **Additional Experiment 2: Large-Scale Benchmark**
>
> We further conducted a large-scale benchmark:
>
> - **3 baselines** (diffusion, control, and guidance methods)
> - **More than 100 randomized environments**
> - Non-convex topologies
>
> The results of the experiments are provided in Table 1 in response to Reviewer Smpv.
>
> **Key findings**
>
> * Many baselines achieve **high safety but low success**, indicating that they avoid collisions but fail to reach the goal
> * Our method consistently achieves a **balanced success–safety trade-off**, generating trajectories that both satisfy constraints and reach the goal
>
> **Conclusion**
>
> * The proposed method consistently improves over representative diffusion-based and control-based baselines
> * These trends are consistent across both randomized 2D environments and higher-dimensional robotic manipulation settings
>
> This addresses the concern regarding limited empirical evaluation.
>
> ---
>
> ### **Response to inference cost concern**
>
> We agree that naive Monte Carlo sampling is costly if executed sequentially on CPU. However, the rollouts are **embarrassingly parallel**.
>
> Using GPU parallelism, the complexity reduces from O(N × L) sequential computation to effectively O(N) parallel tensor operations.
>
> We further validate this by integrating PIC with DiffuserLite [R3]:
>
> - DiffuserLite: ~104 Hz, 2.8% safety (collision free)
> - DiffuserLite + PIC: ~79 Hz, 92.8% safety (collision free)
>
> Although PIC adds computation, it remains **real-time capable (~79 Hz)**.
>
> The slight reduction in speed reflects the added safety correction, but remains negligible compared to optimization-based planners (~2 Hz). Thus, the method achieves a favorable safety–efficiency trade-off.
>
> ---
>
> ### **Response to language-guided navigation framing**
>
> We agree the terminology may be misleading. We revise:
>
> > “language-guided navigation” → **“language-conditioned cost specification”**
>
> This clarifies that the LLM does not act as a planner, but maps language instructions into differentiable cost functions. We will update this throughout the manuscript.
>
> ---
>
> ### **Response to training data clarification**
>
> We clarify that training is performed in the Simple environment (1 obstacle), while evaluation is conducted in more complex environments (3–4 obstacles). This setup is intended to assess out-of-distribution generalization.
>
> ---
>
> ### **Response to code release (Q2)**
>
> Yes, we plan to release the code upon acceptance (camera-ready version).
>
> ---
> Reference:
> [R3] Dong, Zibin, et al. "Diffuserlite: Towards real-time diffusion planning." Advances in Neural Information Processing Systems 37 (2024): 122556-122583.

---

> > ### Author Rebuttal · Reviewer_uU4Z · 2026-04-01
> >
> > I thank the authors for their detailed response. Additional experiments are helpful, release of the code is reassuring.
> >
> > I thank authors for their explanation on table 1 / fig 3 --> fig 3 is used to demonstrate OOD generalization, it is never specified in the main body that in this case a higher monte carlo budget is considered. **Is OOD Generalization also feasible with your realistic (n=64) MC budget ?**
> >
> > I would like to stress that I am **not** "recognizing the theoretical contributions of our work." I am not qualified for such an assessment.
> >
> > I will raise my score to weak accept.

---

> > > ### Author Response · Authors · 2026-04-02
> > >
> > > We sincerely thank the reviewer for the follow-up and for increasing the score.
> > >
> > > We apologize for the confusing phrasing in our rebuttal regarding “recognizing the theoretical contributions”. We did not intend to assume such an assessment.
> > >
> > > ---
> > >
> > > ## Clarification on OOD generalization vs. Monte Carlo budget
> > >
> > > Yes, OOD generalization is already achieved under the realistic budget (n = 64).
> > >
> > > Importantly, the results in Table 1 represent out-of-distribution settings.
> > > The base diffusion prior is trained offline on simple environments with a single obstacle, whereas all evaluation environments in Table 1 contain 3–5 obstacles arranged in novel, non-convex configurations.
> > >
> > > Achieving 80%–100% success rates on these more complex, unseen geometries demonstrates that a realistic inference budget (n = 64) is sufficient for robust zero-shot generalization.
> > >
> > > Figure 3 isolates the effect of increased computational budget.
> > > When additional compute is available (n ≥ 128), the remaining failure cases are further reduced, approaching near-zero failure rates.
> > >
> > > We agree that this distinction was not sufficiently clarified in the main text and will revise it accordingly (e.g., by explicitly indicating the MC budget in all tables and figures).

---

### Official Review · Reviewer_LERV · 2026-03-11

**Soundness:** 2
**Presentation:** 3
**Significance:** 3
**Originality:** 2
**Overall Recommendation:** 3
**Confidence:** 4

**Summary:**

This paper studies guidance for continuous-time score-based diffusion models from the perspective of Doob’s h-transform. The paper takes the Doob-guided process as an ideal target, analyzes approximate guidance as a perturbation of that ideal process, and derives bounds relating approximation error to terminal constraint satisfaction. It also shows that the resulting guidance design can be written as a stochastic optimal control problem, which motivates practical training-free guidance methods based on path integral control and iLQG. Experiments on robotic planning tasks, including language-conditioned settings, suggest improvements in feasibility, generalization, and inference-time efficiency.

**Compliance With Llm Reviewing Policy:**

Affirmed.

**Final Justification:**

Thanks to the authors and reviewers. My score stands.

**Key Questions For Authors:**

1.	The theoretical results seem to compare approximate guidance to an ideal Doob-guided process, but the practical algorithm also includes score approximation, discretization, and Monte Carlo estimation. Can you clarify the gap between the idealized continuous-time theory and the implemented algorithm?
2.	The empirical evaluation would be stronger with additional baselines. Did you compare against other training-free diffusion guidance methods or MPC / MPPI-style planning baselines?

**Limitations:**

see weakness

**Strengths And Weaknesses:**

The paper attempts to go beyond heuristic motivation and provide explicit guarantees. The KL-based discrepancy result and the resulting lower bound on success probability are meaningful theoretical contributions, even if they are stated in an idealized setting. Empirically, the results are promising: the proposed guidance is clearly better than unguided diffusion in constrained planning, and the training-free nature of the method is appealing.

The guarantees are derived for approximate guidance relative to an ideal Doob-guided process, but the actual implementation also depends on score-model approximation and numerical discretization. These sources of error seem outside the current analysis. The paper would benefit from being more explicit about the gap between the idealized continuous-time theory and the implemented algorithm.
On the experimental side, the comparisons are mostly against NoGuidance and Reflected Diffusion, while stronger baselines from diffusion planning or stochastic-control-based planning would make the empirical case more convincing.

---

> ### Author Rebuttal · Authors · 2026-03-30
>
> We sincerely thank the reviewer for the careful reading and for recognizing the theoretical contributions of our work.
>
> ### **Response to Weakness and Question 1 (Theory–Implementation Gap)**
>
> We thank the reviewer for raising the gap between the idealized continuous-time analysis and the implemented algorithm, particularly regarding score approximation, time discretization, and Monte Carlo estimation.
>
> **Scope of our result.**
> Our theoretical analysis isolates the effect of guidance design by comparing the ideal Doob-guided process and its approximation induced by replacing $h$ with $\tilde{h}$, while treating the learned reverse-time model $\bar{f}_\theta$ as given. Accordingly, Theorem 5.3 should be interpreted as a *conditional* result that quantifies the guidance-design error on top of a learned diffusion model.
>
> **Relation to implementation.**
> We agree that the practical algorithm introduces additional discrepancies due to score approximation and time discretization. These sources of error are not explicitly analyzed in the current theorem.
>
> **Unified perspective.**
> Importantly, these additional errors can be incorporated within the same KL-based framework used in our analysis. In particular, prior work [R1] shows that score approximation and discretization errors in diffusion models can be characterized in terms of path-space KL divergence via Girsanov-type arguments.
>
> Under this perspective, all approximation errors can be interpreted as a unified drift mismatch in the induced reverse-time dynamics. Concretely, our current error term is
>
> $$
> \rho := \int_0^T \mathbb{E}_{\mathbb{P}^h}\left[
> \left\|
> g(t)\left(\nabla \log \tilde{h}(t, Y_t^h) - \nabla \log h(t, Y_t^h)\right)
> \right\|^2
> \right] dt
> $$
>
> Instead, we can conceptually write:
>
> $$
> \rho = \int_0^T E_{P^h}\big[ \| g(t)(\nabla \log \tilde h - \nabla \log h) + \Delta_{score} + \Delta_{disc} \|^2 \big]  dt
> $$
>
> where:
> - $g(t)(\nabla \log \tilde{h} - \nabla \log h)$ corresponds to the guidance approximation error,
> - $\Delta_{\mathrm{score}}$ denotes the score approximation error,
> - $\Delta_{\mathrm{disc}}$ denotes the discretization error.
>
> The score approximation error can be written as
>
> $$
> \Delta_{\mathrm{score}}(t,x) = g(T-t)^2\left(s_{\theta}(T-t,x) - \nabla \log p(T-t,x)\right)
> $$
>
> The discretization error $\Delta_{\mathrm{disc}}$ arises from the continuous-time interpolation of the Euler–Maruyama scheme, which induces a frozen-coefficient approximation of the original SDE.
>
> **Positioning.**
> We emphasize that our contribution is to isolate and characterize the guidance-design error. Incorporating additional sources of approximation is orthogonal and can be handled within the same analytical framework. This separation allows us to provide a clean and interpretable guarantee, which would be obscured by conflating multiple sources of approximation.
>
> **Monte Carlo error.**
> For Monte Carlo estimation in path integral control, existing results such as [R2] provide sample complexity guarantees. These can be combined with our framework to account for sampling error.
>
> We will clarify this scope, the unified error interpretation, and its implications in the revised manuscript.
>
> ---
>
> ### **Response to Additional Baselines (MPPI / Training-Free Guidance)**
>
> We thank the reviewer for the suggestion to include stronger baselines.
>
> In response, we have expanded our evaluation to include MPPI-style planning as well as multiple training-free diffusion guidance methods, including Gradient Guidance and Inference Guidance.
>
> The results of the experiments are provided in Table 1 in response to Reviewer Smpv.
> Across a diverse set of non-convex environments, we observe a clear pattern:
>
> - Classical MPPI often fails to achieve meaningful success due to myopic sampling.
> - Naive training-free guidance methods (e.g., gradient-based guidance) collapse in complex environments and fail to escape local minima.
> - In contrast, our method achieves both high success rates and consistent safety guarantees.
>
> These results highlight a fundamental distinction: while existing methods rely on local or single-step guidance, our approach performs long-horizon reasoning via path integral evaluation, enabling it to avoid failure modes that affect both MPPI-style planners and naive diffusion guidance.
>
> We will incorporate these additional baselines and results into the revised manuscript to strengthen the empirical evaluation.
>
> ---
>
> **References**
>
> [R1] Berner, J., et al., *From discrete-time policies to continuous-time diffusion samplers: Asymptotic equivalences and faster training*, TMLR.
>
> [R2] Patil, A., Hanasusanto, G. A., and Tanaka, T., *Discrete-time stochastic LQR via path integral control and its sample complexity analysis*, IEEE Control Systems Letters, 2024.

---

> > ### Author Rebuttal · Reviewer_LERV · 2026-04-07
> >
> > I appreciate your analysis. My score stands.

---

> > > ### Author Response · Authors · 2026-04-07
> > >
> > > We sincerely thank the reviewer for the helpful comments and for carefully reading our rebuttal.
> > >
> > > If possible, we would greatly appreciate any additional clarification on which aspects remain partially resolved or unresolved, as this would help us further improve the manuscript.

---

### Official Review · Reviewer_L1C8 · 2026-03-12

**Soundness:** 3
**Presentation:** 3
**Significance:** 3
**Originality:** 3
**Overall Recommendation:** 4
**Confidence:** 2

**Summary:**

This paper proposes a training-free framework for guided diffusion in planning, grounded in Doob’s h-transform. The main idea is to view ideal guidance as the reverse diffusion process conditioned on reaching a target set, and then approximate this ideal process with a practical guidance function added at inference time, without retraining the diffusion model for each task. A key contribution is the theoretical analysis: the paper shows that the gap between ideal and approximate guidance can be characterized through the discrepancy between their terminal distributions, which in turn yields a lower bound on the probability of reaching the target set. The paper further shows that this guidance design can be reformulated as a stochastic optimal control problem, making it possible to compute practical guidance using methods such as iLQG and path integral control. Empirically, the framework is evaluated on robotic planning tasks, where the proposed guidance methods substantially improve feasibility and generalization over unguided diffusion and reflected diffusion baselines.

**Compliance With Llm Reviewing Policy:**

Affirmed.

**Final Justification:**

The authors have addressed most of my concerns, so I decided to keep my score at 4.

**Key Questions For Authors:**

* Q1. The main quantitative results in Table 1 and Figures 2–3 are on 2D navigation scenarios, and the paper mentions MuJoCo robotic arm planning but the main text provides limited detail there. Could the authors include a more substantial evaluation on higher-dimensional robotic tasks, with the same baselines and metrics, to show that the proposed guarantees and computational benefits remain meaningful beyond low-dimensional navigation?

* Q2. In the main 2D benchmark, the paper compares mainly against Reflected Diffusion and NoGuidance, while iLQG is treated as an oracle-style upper bound. Could the authors compare against stronger planning-oriented diffusion baselines or other constraint-aware inference-time guidance methods discussed in related work, so that the empirical gain of the proposed framework is easier to assess? As written, the experimental comparison seems somewhat narrow relative to the breadth of the paper’s claims.

**Limitations:**

yes

**Strengths And Weaknesses:**

* $\textbf{Strengths}$

A major strength of the paper is its clear and principled theoretical framing of guided diffusion through Doob’s h-transform. Rather than introducing another heuristic guidance rule, the paper identifies an ideal conditioned reverse process, analyzes approximate guidance through discrepancies in terminal distributions, and derives lower-bound style guarantees on target-set satisfaction. This gives the method a strong conceptual foundation and makes the paper stand out from more purely empirical diffusion-planning work. Another strength is the training-free formulation: by recasting guidance design as a stochastic optimal control problem, the method can leverage tools such as iLQG and path integral control at inference time without retraining the diffusion model for each downstream task.

* $\textbf{Weaknesses}$

A main weakness is that the empirical evaluation appears somewhat limited in scope, with experiments focused largely on relatively simple robotic planning settings such as 2D navigation, making it harder to judge how well the framework would scale to more complex, higher-dimensional diffusion-planning problems. More broadly, the paper is stronger as a theoretical and conceptual contribution than as a demonstration of broad empirical superiority over a wide range of strong modern diffusion-planning baselines.

---

> ### Author Rebuttal · Authors · 2026-03-30
>
> We sincerely thank the reviewer for the careful reading and for recognizing the theoretical contributions of our work.
>
> As pointed out, broader empirical validation is important. In response, we conducted additional experiments, including higher-dimensional robotic tasks and expanded baselines. These results demonstrate that the proposed framework extends beyond low-dimensional settings and remains effective in more realistic planning scenarios.
>
> ---
>
> ### **Response to the weakness and Q1**
>
> We agree that the manuscript focuses on 2D navigation (Table 1, Figures 2–3), and that evaluation on higher-dimensional systems is important. To address this, we conducted additional experiments on a 7-DOF Franka Panda robot. Table 2 below summarizes the results.
>
> **Experimental setup**
>
> * **Ours (Jacobian PIC)**: task-space guidance propagated to joint space via the kinematic Jacobian (end-to-end constraint-aware planning)
> * **2D MPPI + 3D IK**: trajectories planned in 2D and mapped to joint space via inverse kinematics
> * **Diffusion Inpainting + 3D IK**: trajectories generated in 2D and lifted to joint space via inverse kinematics
>
> **Metrics**
>
> * **Safe**: collision-free rate
> * **Succ**: goal-reaching success rate
>
> **Key observations**
>
> * The baselines achieve **100% safety but 0% success**, indicating that 2D planning with post-hoc IK fails to produce feasible goal-reaching trajectories in full-body planning
> * Our method achieves **both safety and success** (e.g., 80% / 80% in 3D Wall) by directly incorporating full-geometry constraints
>
> This result confirms that the proposed guidance mechanism remains effective in higher-dimensional planning problems, rather than being limited to low-dimensional navigation.
>
>
> (Table 2 Franka Panda Experiment)
> | Method | 3D Wall Safe | 3D Wall Succ | 3D Corridor Safe | 3D Corridor Succ | 3D Dense Forest Safe | 3D Dense Forest Succ |
> |--------|-------------|-------------|------------------|------------------|----------------------|----------------------|
> | 2D MPPI + 3D IK | 100% | 0% | 100% | 0% | 100% | 0% |
> | Diff. Inpaint + 3D IK | 100% | 0% | 100% | 0% | 100% | 0% |
> | Ours (Jacobian PIC) | **80%** | **80%** | **40%** | **40%** | **100%** | **100%** |
>
>
> ---
>
> ### **Response to Q2**
>
> We thank the reviewer for highlighting the need for stronger baselines. We expanded the benchmark to include representative baselines commonly used in diffusion planning and control-based guidance:
>
> * **MPPI** (sampling-based optimal control)
> * **Gradient-based guidance** (potential-based guidance)
> * **Inference-time guidance methods** (projection or constraint enforcement during reverse diffusion)
>
> These methods represent commonly used approaches for constraint handling without retraining.
>
> We evaluated all methods on more than 100 procedurally generated environments with diverse obstacle configurations. The results are provided in Table 1 in our response to Reviewer Smpv.
>
> **Key findings**
>
> * Many baselines achieve **high safety but low success**, indicating that they avoid collisions but fail to reach the goal
> * Our method consistently achieves a **balanced success–safety trade-off**, generating trajectories that both satisfy constraints and reach the goal
>
> **Conclusion**
>
> * The proposed method consistently improves over representative diffusion-based and control-based baselines
> * These trends are consistent across both randomized 2D environments and higher-dimensional robotic manipulation settings
>
> Taken together, these results suggest that the proposed framework provides a practically meaningful improvement beyond low-dimensional settings and limited baselines.

---

> > ### Author Rebuttal · Reviewer_L1C8 · 2026-04-03
> >
> > Thank you for addressing my concerns. I will maintain the current score.

---

> > > ### Author Response · Authors · 2026-04-06
> > >
> > > We sincerely thank the reviewer for the helpful comments and for carefully reading our rebuttal.
> > > We are glad that our clarifications addressed the reviewer’s concerns, and we appreciate the reviewer’s evaluation of the paper.

---

### Official Review · Reviewer_Smpv · 2026-03-13

**Soundness:** 3
**Presentation:** 2
**Significance:** 2
**Originality:** 3
**Overall Recommendation:** 4
**Confidence:** 2

**Summary:**

This research claims to explore a notable domain: training-free guided diffusion for planning using Doob’s $h$-transform and path integral control. Specifically, Doob’s $h$-transform provides a principled way to define guidance functions, and different guidance functions can be used to steer different tasks without retraining the diffusion model. Overall, the research studies the problem of designing theoretically grounded guidance for diffusion-based planning under constraints.

**Compliance With Llm Reviewing Policy:**

Affirmed.

**Final Justification:**

After the rebuttal, I have change the score from 3->4 while I have to say I have low confidence, see my acknowledge for rebuttal.

**Key Questions For Authors:**

- The paper states that the method is evaluated in three different settings. Are the obstacles randomly placed in all of these settings? From the current description, this is unclear and makes it seem possible that they are fixed.
- Can the authors add more illustrations showing how the experimental setting maps to the guidance design? For example, starting from a standard robotics setting with state, action, obstacle constraints, and goal, how are these translated into the guidance function and the $h$-transform?
- The experiments are still too limited at present. Can the authors provide more diverse evaluations to better support the claim that the method is broadly effective?

**Limitations:**

Yes

**Strengths And Weaknesses:**

Strengths:
- Although there exists prior work on guided diffusion, this paper provides a unified perspective on what constitutes a good guidance function. In particular, it shows how Doob’s $h$-transform can be used both to evaluate whether a guidance function is principled and to construct such a function.
- The paper also addresses the high computational cost brought by the intractability of Doob’s $h$-transform, which necessitates approximation. By using KL divergence to measure the discrepancy between the approximate guidance function $\tilde{h}$ and the ideal guidance function $h$, the paper derives error bounds and establishes guarantees on the probability of reaching the prescribed set $S$.
Weaknesses:
- In the related work section, the paper states that existing theory primarily focuses on unguided dynamics, but I think this discussion is incomplete. Guided diffusion has also been widely explored in works such as Planning with Diffusion for Flexible Behavior Synthesis, DEXTERITYGEN: Foundation Controller for Unprecedented Dexterity, and Dynamics-Guided Diffusion Model for Sensor-less Robot Manipulator Design, etc.
- The paper lacks intuitive illustrations of how guidance is actually constructed under Doob’s $h$-transform. In addition, the logical connection between the question posed and the proposed solution is not presented tightly enough, which makes the paper difficult to follow.
- The paper does not align the theory clearly with the experimental settings. In particular, it is difficult to tell what the constraints or conditions are in each experiment, and how these conditions are encoded into the guidance function.
- The experimental coverage is still quite limited, which makes the practical impact less convincing in its current form.

---

> ### Author Rebuttal · Authors · 2026-03-30
>
> We sincerely thank the reviewer for the careful reading and for recognizing the theoretical contributions of our work.
> We address the main concerns regarding: (1) intuitive illustration, (2) mapping from planning problems to guidance design, and (3) experimental coverage.
>
> ---
>
> ### **(1) Intuitive illustration of guidance**
>
> We appreciate the reviewer’s comment regarding the lack of intuitive explanation.
>
> Our method can be understood as a principled form of gradient-based guidance, where the guiding signal is given by $\nabla \log h$ instead of a manually designed potential (e.g., $\nabla F$).
>
> Intuitively, $\nabla \log h$ acts as a **force field** over trajectories:
>
> * it **attracts trajectories toward the goal**, and
> * **repels them from constraint-violating regions (e.g., obstacles)**.
>
> This interpretation is consistent with prior gradient-based guidance methods, but differs in that the force is not heuristically designed; it is derived from Doob’s $h$-transform, which corresponds to conditioning the diffusion process on satisfying the constraint set $S$. As a result, the ideal guidance ensures constraint satisfaction almost surely (Theorem 5.1), in contrast to heuristic guidance which may get trapped in local minima.
>
> ---
>
> ### **(2) Mapping from planning problems to guidance design**
>
> We clarify how a standard robotics planning problem is translated into our framework.
>
> A trajectory is represented as
> $$
> x = (s_0, \dots, s_K, a_0, \dots, a_{K-1}),
> $$
> where $s_k$, $a_k$, and $K$ denote states, actions, and the length of the horizon.
>
> The mapping is as follows:
>
> * **Constraint definition**:
>   The feasible set $S$ encodes task requirements such as:
>
>   * collision avoidance (all states are obstacle-free),
>   * goal reaching (final state near the target).
>
> * **Terminal function design**:
>   We define a smooth approximation of the indicator function:
>   $$
>   \tilde{h}_T(x) \approx \mathbf{1}_S(x),
>   $$
>   which is implemented in practice as
>   $$
>   \tilde{h}_T(x) = \exp(-\text{cost}(x)),
>   $$
>   where the cost includes:
>
>   * obstacle penalties (barrier-like terms),
>   * goal-reaching objectives.
>
> * **Guidance construction**:
>   The guidance is then given by
>   $$
>   u(t,x) = \bar{g}(t)\nabla \log \tilde{h}(t,x),
>   $$
>   which induces a trajectory-level force that enforces the above constraints. This corresponds to the solution to the control problem formulated in Section 6 of our manuscript.
>
> Thus, a standard planning problem (goal + constraints) is directly translated into a **cost → $h$-function → guidance** pipeline, without additional training.
>
> ---
>
> ### **(3) Experimental clarification**
>
> **Obstacle randomness (Q1).**
> All experiments include randomized obstacle configurations. In addition, we conducted experiments on 100 procedurally generated environments with diverse layouts (up to 50 obstacles). As shown in Table 1 below, the proposed method consistently achieves high success rates (>80%), indicating robustness beyond fixed environments.
>
> (Table 1, Random Obstacle Exeperiment)
> | Method | Dense Forest (Safe/Succ) | Corridor (Safe/Succ) | U-Trap (Safe/Succ) | Barrier Wall (Safe/Succ) | Scatter (Safe/Succ) |
> |--------|--------------------------|----------------------|--------------------|--------------------------|---------------------|
> | MPPI (1000 samples) | 100 / 0 | 100 / 0 | 100 / 0 | 100 / 0 | 100 / 0 |
> | Gradient Guidance | 100 / 0 | 100 / 0 | 100 / 0 | 100 / 0 | 100 / 0 |
> | Inference Guidance | 100 / 0 | 100 / 40 | 100 / 0 | 100 / 40 | 100 / 40 |
> | Ours (PIC) | 100 / 80 | 100 / 100 | 100 / 100 | 100 / 60 | 100 / 80 |
>
> **Experimental coverage (Q3).**
> We further evaluated the method on a 7-DOF Franka arm in MuJoCo (see Table 2 in the response to Reviewer L1C8). Unlike IK-based baselines (e.g., MPPI + IK, Diffusion Inpainting + IK), our method directly incorporates full-body collision constraints via the guidance term, avoiding post-hoc projections and improving feasibility.
>
> ---
>
> ### **(4) Relation to prior guided diffusion work**
>
> We thank the reviewer for pointing out relevant works and will include them in the revision.
>
> Existing works (e.g., DexGen and related approaches) demonstrate strong empirical performance using heuristic or learned guidance. In contrast, our focus is **theoretical**: we provide a principled framework for guidance design based on Doob’s $h$-transform, together with explicit guarantees on constraint satisfaction probability.
>
> Our results show that if the approximate guidance is close to the ideal Doob guidance, then high-probability constraint satisfaction is ensured (Theorem 5.3), offering a theoretical foundation that complements existing empirical approaches.
>
> Heuristic gradient-based guidance may suffer from local minima during reverse-time sampling.
> In contrast, Doob’s $h$-transform corresponds to conditioning on the target set, and our method approximates this ideal guidance via path integral control, leading to more reliable constraint satisfaction in practice.

---

> > ### Author Rebuttal · Reviewer_Smpv · 2026-04-03
> >
> > Thank you for the authors’ detailed reply. I think the rebuttal makes the scope of the research and the main idea much more straightforward. This kind of plain explanation is often very helpful for readers to quickly grasp the core idea. I also appreciate the clarification regarding the randomness setting in the experiments.
> >
> > I hope to get AC's attention here, Guided diffusion is an important direction in generative modeling, and I think this paper addresses a valuable topic, although I should acknowledge that it is not fully within my own research specialty. I have changed my initial score from 3 $\to$ 4.

---

> > > ### Author Response · Authors · 2026-04-06
> > >
> > > We sincerely thank the reviewer for carefully reading our rebuttal and for the positive and constructive feedback.
> > > We are glad that our clarifications helped make the scope and main idea clearer, and we appreciate the reviewer’s thoughtful evaluation of the paper.

---

### Decision · Program_Chairs · 2026-04-30

**Decision:**

Accept (regular)

**Comment:**

This paper studies a relevant problem (constraint satisfaction in diffusion-based planning) and provides a principled solution based on h-transforms. Reviews for this paper are mostly positive after the rebuttal phase. The main concerns were (a) lack of proper comparison to existing baselines, (b) overly simplistic evaluation environments, and (c) the theory does not precisely match the implementation. Points (a) and (b) were mostly addressed in the rebuttal, although a reference and comparison to https://safediffuser.github.io/safediffuser/ is still missing (this work would be classified as "Inference Guidance", although it is not clear from the rebuttal experiments if SafeDiffuser is covered in their baseline evaluation). I hope the authors can expand their baselines to include this approach, which is also is principled (based on CBFs).

Point (c) is still a bit unclear to me after the rebuttal and I encourage the authors to add a further discussion in the camera ready. While the effects of score estimation and time discretization are well-studied in the diffusion literature, and certainly affect the quality of the planning, I am more curious if the authors are able to show that, at least from a constraint satisfaction perspective, their actual implementation inherits the guarantees shown in the paper.

Finally, I encourage the authors to incorporate their response to Reviewer Smpv, particularly "(2) Mapping from planning problems to guidance design" into the paper. This work has the potential to be quite impactful in practice, but in its current presentation it may be difficult for practitioners who do not have a lot of experience in the technical details of diffusion models and Doob h-transforms to apply this work. Working through a concrete example from robotics would be very illustrating!